# Temporal predictions shape somatosensory perception

Andreas Strube [1,2] ✉ & Christian Büchel [1] ✉

Although intensity expectations have been thoroughly studied in relation to pain, there has been a notable lack of investigation into temporal expectations. One important temporal pain effect, the so-called dread effect, suggests that future pain becomes more aversive with increasing delay. Here we investigated temporal expectations including the dread effect by presenting probabilistically cued painful heat and non-painful cold stimuli after different delay periods. Actual stimulus latency had no effect on perceived intensity in both non-painful cold and painful heat conditions. However, our data clearly show that the expectation of longer delays amplified somatosensory perception, indicating that the dread effect is related to expected and not to experienced delay. Electroencephalography data show that temporal expectations modulate alpha/beta activity during cue presentation, but not during stimulation. Actual stimulus timing is represented in alpha-to-beta frequencies during heat and cold stimulation.

Whenever we wait for a painful event, we generate expectations regarding the timing of the painful stimulus, making this a possible example of predictive coding. Predictive coding posits that stimulus expectations, such as intensity and timing, are fundamental to perception[1–3]. From the lens of this framework, we consistently build a model of our world and compare it with actual sensory input. Mismatches between this model and the actual sensory information would result in prediction errors, which are utilized to form more accurate future models of the world. This mechanism would allow us to reduce the amount of processed information, which in turn leads to more efficient neural processing, and, in the context of the Free Energy Principle[3–5], to a reduction in surprise/entropy. The recurrent nature of pain-related neural activity has led to the suggestion that pain processing can be investigated through the lens of predictive processing[5–13].

Previously, pain processing has been shown to reflect the interaction of sensory input properties, expectations, and prediction errors[7,12,14–20]. However, these studies have primarily focused on demonstrating these effects within the context of predicting pain intensity. For example, if someone expects to feel low pain but suddenly experiences high pain instead, this creates a prediction error.

This line of reasoning has led to the formulation of a Bayesian integration model of pain perception[6,13,21–23]. According to this model, precision-weighted prior expectations are combined with sensory interoceptive information to generate an individual's pain perception. This has been shown in many studies investigating the relationship of pain and pain intensity expectations[7,12,15,24–31].

In contrast, few studies have investigated the properties of temporal expectations regarding pain, which raises the question of how temporal sensory input, expectations, and prediction errors are processed, for example, what if pain occurs earlier than expected? This specific form of predictive timing remains largely unexplored, highlighting the need for further investigation into these mechanisms to enhance our understanding of how temporal expectations influence pain processing.

One notable phenomenon related to timing in pain perception is the "dread effect," which challenged traditional economic theories of temporal discounting[32,33]. In classical decision-making theories, it was posited that individuals tend to defer the experience of negative consequences while prioritizing the pursuit of positive outcomes[34,35]. However, these theories do not consider the possibility that anticipation itself can have an

[1]Department of Systems Neuroscience, University Medical Center Hamburg-Eppendorf, Hamburg, Germany. [2]Present address: Center for Depression, Anxiety and Stress Research, Department of Psychiatry, McLean Hospital, Harvard Medical School, Boston, MA, USA. ✉e-mail: mail@andreasstrube.de; buechel@uke.de

appetitive or aversive value. On the one hand, we experience a certain sense of joyful anticipation for a distant vacation, and this anticipation has intrinsic value. On the other hand, we experience a certain sense of dread in anticipation of a dentist appointment (as has been shown by Story et al.[36]). This clearly affects decision making: When presented with a decision regarding the waiting time for potential pain, most individuals tend to choose an earlier rather than a later outcome[36,37]. This phenomenon is known as the "dread effect", as anticipation itself seems to be associated with negative effects of dread. Similar to this notion, longer anticipation periods have been linked to an increased perception of pain[38,39], and are particularly relevant to investigate within the framework of predictive timing, as it may reflect how temporal expectations shape processing of pain during a waiting period. However, none of these studies have distinguished between an expected waiting period and the actual waiting period.

Building upon prior work demonstrating that pain perception is modulated by probabilistic expectations and prediction errors[7,12,15,16], our study investigated how probabilistic temporal expectations and actual delays influence pain perception and associated brain (EEG) activity.

We designed an EEG experiment (adapted from Strube et al.[12,40]), in which we could induce temporal expectations (instant, early, late) regarding a somatosensory non-painful cold or painful heat stimulus (see Fig. 1 for an overview and Fig. 2 for the trial design). We deliberately included non-painful cold stimuli to ascertain whether these effects stem from general somatosensory processing or whether they are specific to painful outcomes. We aimed to specifically distinguish between an expected waiting period and an actual waiting period. This approach allowed us to explore several key questions: First, can we identify a distinct pattern in brain activity for predictive timing, i.e., temporal expectations and prediction errors? Second, what is the potential role of timing expectations versus actual delays with respect to the dread effect[36,37]?

## Results

In this experiment, we initially presented a cue indicating whether the subsequent stimulus would result in an increase to (painful) 46.5 °C heat or a decrease to (non-painful) 20.5 °C cold from a baseline temperature of 30 °C. A second cue indicated, probabilistically, if the next (painful) heat stimulus or (non-painful) cold stimulus started instantly, early or late (0 s, 2 s, and 4 s after presentation of the temporal cue, respectively), inducing temporal expectations. After stimulation, participants were asked to rate the stimulus on a visual analog intensity scale (VAS), where 0–50 represented non-painful intensities, and 51–100 represented painful intensities.

### Behavioral intensity ratings

We analyzed trial-wise intensity ratings using a linear mixed-effects (LME) model with fixed effects of stimulus modality (heat vs. cold), temporal expectation (i.e., expected latency based on predictive cues), stimulus latency (i.e., actual latency from cue-offset to stimulation-onset), and prediction error (i.e., absolute difference between expectation and latency), including all interactions with stimulus modality. The model included by-participant random intercepts and random slopes for all predictors. See "Methods" for a detailed description of the LME model (Fig. 1b). See Supplementary Information for full model output and diagnostics.

The model revealed a significant main effect of stimulus modality ($t(10072) = 16.88$, $p < .001$, standardized $\beta = 1.69$, SE = 0.10, 95% CI [1.50, 1.89]), i.e., (non-painful) cold stimuli were rated with lower intensities than (painful) heat (see Fig. 3a, b for heat and cold ratings). In addition, we observed a significant main effect of temporal expectations ($t(10072) = 3.22$, $p = .001$, standardized $\beta = 0.02$, SE = 0.006, 95% CI [0.008, 0.033]), indicating that intensity ratings increased with longer expected latency.

In contrast, the actual latency between cue and stimulation did not significantly modulate intensity ratings ($t(10072) = 0.14$, $p = .886$, standardized $\beta < 0.001$, SE = 0.006, 95% CI [−0.011, 0.013]), i.e., longer

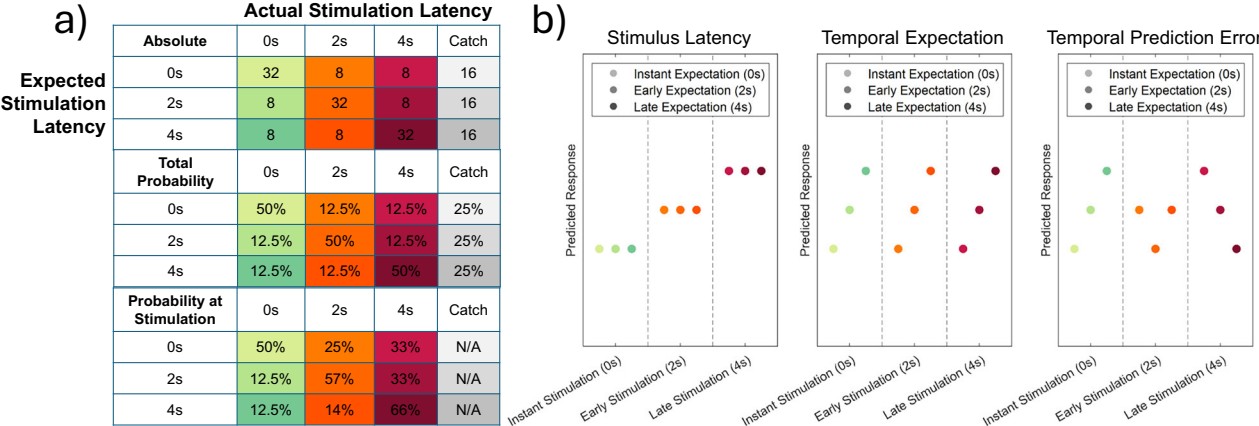

**Fig. 1 | The predictive timing paradigm.** Temporal expectations (instant 0 s, early 2 s, late 4 s) are systematically manipulated to explore how the brain processes expectations, evidence, and prediction errors and adjusts to timing discrepancies during somatosensory processing. **a** Contingency table (equivalent for pain and cold stimulation) for expected and actual stimulation time-points, showing absolute, relative, and probabilities at stimulation onset for each combination of expected and actual stimulation. Absolute numbers indicate the total number of trials associated with each combination of expected and actual delay across the whole experiment. Relative probabilities indicate the probability of a certain expected and actual stimulation combination, given the expected stimulation. Probability at stimulation takes the temporal progression during the trial into account and indicates the probability of stimulation at the onset of stimulation, given the expected stimulation. For example, if stimulation did not occur instantly after 0 s, this changes the probabilities of stimulation at later time points (2 s or 4 s) as the trial temporally progresses. We evaluated the temporal progression to ensure (by including catch trials), for example, that late stimulation (at 4 s) is still improbable for instant and early expectation (0 s and 2 s) conditions. **b** The predictive timing model: Hypothetical response patterns based on stimulus latency (left), temporal expectations (middle), and absolute temporal prediction errors (right). The y-axis represents a hypothetical response variable (e.g., pain intensity or EEG power). Each dot represents a different condition (stimulus-cue combination). Green colors represent instant stimulation conditions (0 s), orange colors represent early stimulation conditions (2 s), and red colors represent late stimulation conditions (4 s). Color intensities depict expected stimulation latency, i.e., more intense colors represent longer expected latencies (0 s, 2 s, and 4 s, respectively).

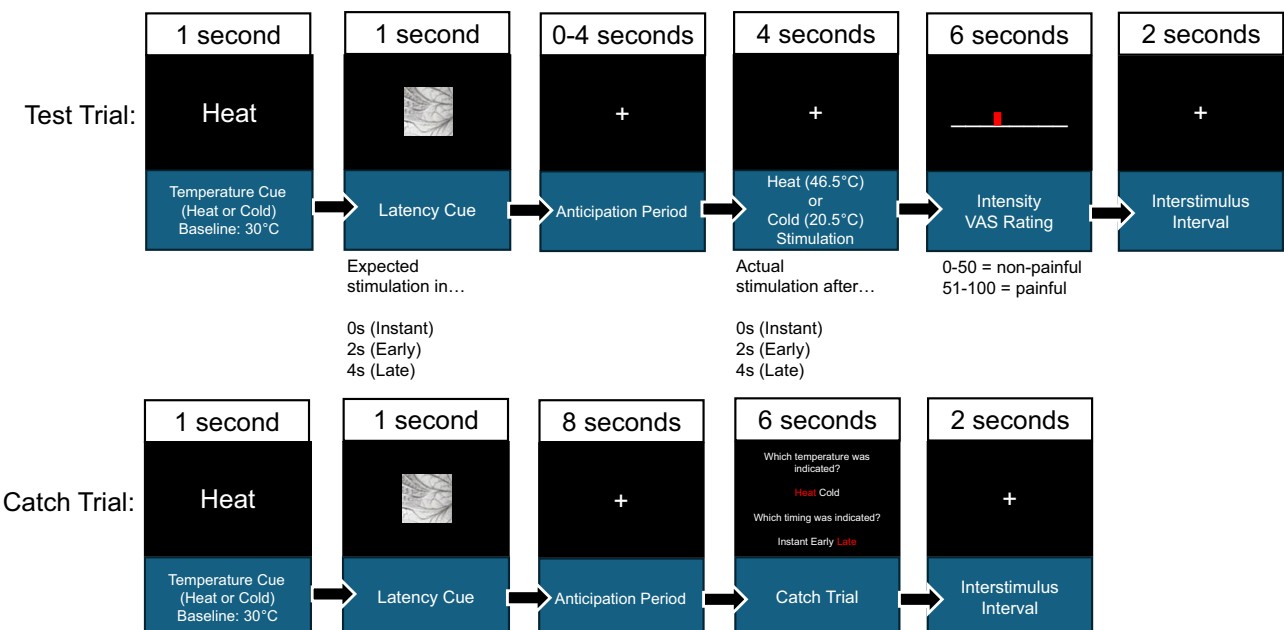

Fig. 2 | **Trial design.** Subjects received either (painful) heat or (non-painful) cold stimuli applied by a thermode on the left forearm, as indicated by a temperature cue (presented for 1 s). Directly after the offset of the temperature cue, a second latency cue (presented for 1 s) induced temporal expectations by probabilistically indicating via (learned) fractal cues whether the interval between the offset of the latency cue and the stimulus was 0 s, 2 s, or 4 s, indicating instant, early, or late stimulation, respectively. In test trials (top), after an anticipation period of 0 s, 2 s, or 4 s, a (painful) heat or (non-painful) cold stimulus was delivered by increasing the baseline temperature from 30 to 46.5 °C or decreasing it to 20.5 °C for 4 s. Afterwards, the stimulus intensity was rated on a visual analog scale (VAS) from 0 to 100, where 0–50 represents non-painful intensities and 51–100 represents painful intensities. In catch trials (bottom), no stimulation was delivered after the maximum anticipation period of 4 s, and participants were asked to indicate which temperature and latency were indicated by the temperature and latency cues.

latencies were not associated with differences in ratings. We also did not observe a significant prediction error effect, i.e., the absolute difference between temporal expectations and stimulus latency did not significantly modulate intensity ratings (t(10072) = −0.20, $p$ = .843, standardized β = −0.001, SE = 0.005, 95% CI [−0.011, 0.009]). Furthermore, no significant interactions of stimulus modality with latency, expectation, or prediction error were observed (all $p$ > .66).

For painful heat stimulation (Fig. 3a), post hoc analyses using Bonferroni correction for multiple comparisons indicated that late temporal expectations lead to higher intensity VAS ratings (M = 72.16, SD = 11.66) than early latency expectations (M = 70.55, SD = 11.79, $p$ < .001) and instant temporal expectations (M = 70.85, SD = 11.74; all $p$ < .001). The difference between instant and early temporal expectations was not significant ($p$ = .94).

In (non-painful) cold stimulation (Fig. 3b), post hoc analyses using Bonferroni correction for multiple comparisons indicated that late temporal expectations lead to higher VAS intensity ratings (M = 22.14, SD = 11.76) than instant temporal expectations (M = 20.67, SD = 11.05, $p$ < .001). The difference between both, early (M = 21.29, SD = 11.03) and instant temporal expectations ($p$ = .25) and early and late temporal expectations ($p$ = .22) were not significant.

An additional supplementary analysis, including prediction errors at stimulation (see Fig. 1a, bottom row), also showed that the data are modulated by expected latency, but not by prediction errors or actual stimulus latency (see Supplementary Information and Supplementary Fig. 2).

This suggests that temporal expectations alone can modulate somatosensory perception, increasing its perceived intensity. This was observed for painful heat and non-painful cold stimuli, i.e., more intense heat and more intense cold perception. Our rating scale allowed us to dissociate painful (ratings between 51 and 100) from non-painful stimulation (ratings between 0 and 50). Heat stimuli ratings (M = 71.19, SD = 11.69) were clearly in the painful range, whereas cold

stimuli (M = 21.37, SD = 11.21) were clearly in the non-painful range. Therefore, the observed effect of temporal expectations applies regardless of the painfulness of the stimulation. This is at odds with interpretations explaining increased pain by longer anticipation periods; instead, this suggests that the somatosensory expectation itself can drive this effect in both painful heat and non-painful cold conditions.

We additionally performed exploratory computational modeling within a Bayesian Pain Model framework[6,30,41–44] to illustrate possible mechanisms behind the expected-delay-dependent intensity increase. Given a scenario in which intensity expectations during acute pain are below the sensory input, model evidence favored an expectation intensity shift model over an expectation precision modulation model. In all tested configurations, expectation intensity shift models outperformed expectation precision modulation models (i.e., protected exceedance probability was at φ ≈ 1). Modeling details are reported in the Supplementary Information.

### EEG results
For the time-frequency analysis of EEG data, we considered two separate time intervals to evaluate cue-locked and stimulus-locked effects. For cue-locked analyses, we set t = 0 to the onset of the cue, probabilistically indicating the latency of the upcoming stimulus. In stimulus-locked analyses t = 0 was set to the point when the thermode was triggered to increase/decrease the temperature to the target levels of 46.5 °C for painful heat and to 20.5 °C for non-painful cold. All tests were corrected for multiple comparisons using Monte Carlo cluster tests. At each sample an F-test was conducted for each respective contrast (i.e., heat vs cold stimulation, stimulation vs catch, stimulus latency, temporal expectations and prediction errors), and all samples exceeding the threshold of $p$ < 0.05 were clustered in connected sets on the basis of temporal (i.e., adjacent time intervals), spatial (i.e., neighboring electrodes), and spectral adjacency. This was adjusted

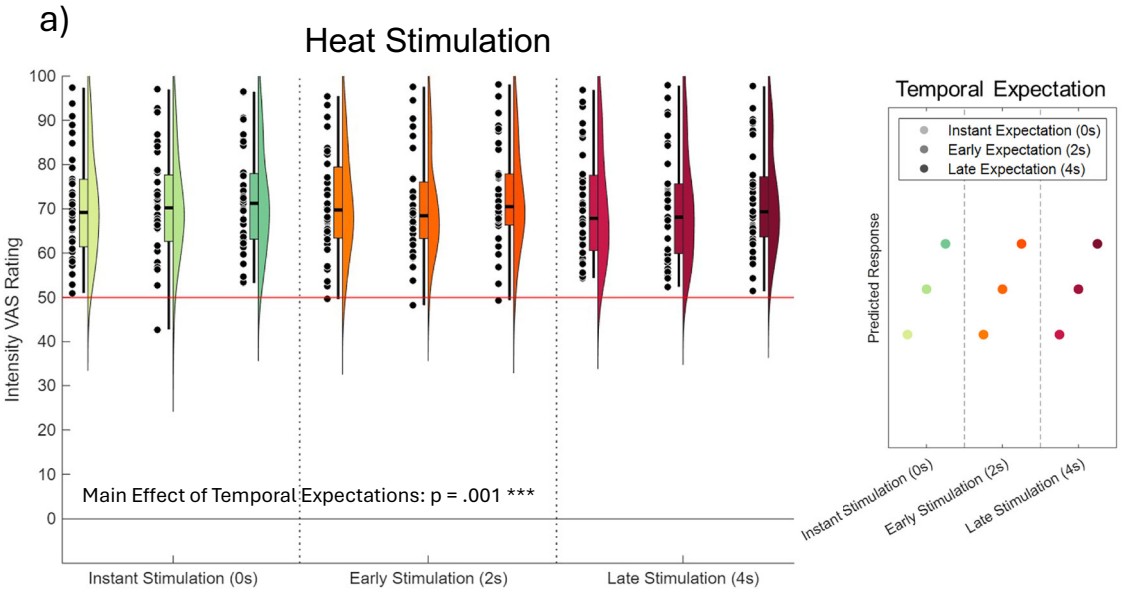

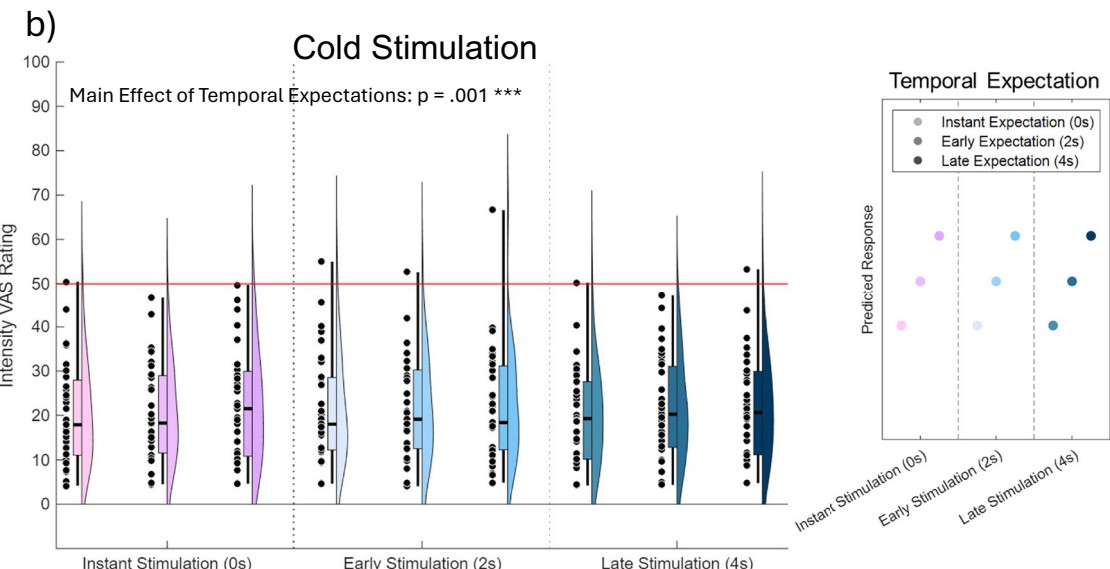

**Fig. 3 | Intensity ratings and temporal expectations.** Somatosensory (non-painful) cold and (painful) heat intensity ratings are modulated by temporal expectations. **a**, **b** show ratings for heat (painful, 51–100 VAS, warm colors) and cold (non-painful, 0–50 VAS, cold colors) stimuli, respectively. Violin plots display the distribution of Visual Analog Scale (VAS) ratings across expectation conditions, overlaid with boxplots indicating mean, interquartile range, and whiskers representing the full data range (min to max). Each violin represents a cue-stimulus condition; color intensity reflects expectation level (low to high expected cue-stimulus delay). Individual subject averages per condition ($n = 35$) are shown as scatter points, illustrating the variability across subjects. Statistical analyses were performed using linear mixed-effects models (two-sided tests), with subjects as random effects. Exact $p$ values, effect sizes, confidence intervals, and degrees of freedom are reported in the main text. Asterisks indicate significant main effects of expectation (*** indicates $p \leq .001$). Ratings were given on a scale from 1 to 100, where 1–50 represent non-painful and 51–100 painful intensities. The panel on the right in each subplot illustrates the theoretical shape of the temporal expectation effect, with the y-axis denoting an arbitrary response variable (e.g., VAS rating), and each dot representing a distinct cue-stimulus condition. This figure was created using the function *daviolinplot* of the MATLAB toolbox DataViz (v3.2.4)[57].

using permutation testing. (see "Methods" for details; see Supplementary Fig. 7 for z-scored EEG power at pain stimulation conditions; see Supplementary Fig. 8 for z-scored EEG power at cold stimulation conditions).

### Effect of pain and cold stimulation
In the first step, we investigated the EEG profile related to pain stimulation to check whether this follows canonical EEG patterns, namely an increase in theta power (4–8 Hz), a decrease in alpha-to-beta power (8–30 Hz), and an increase in gamma power (>30 Hz; see Ploner et al. for review)[45].

Firstly, we conducted an F-test to assess the main effect of modality on EEG power by contrasting (correctly cued, to avoid the influence of prediction errors) non-painful cold and painful heat trials and identified clusters showing significant differences between cold and pain using a Monte Carlo cluster-based permutation approach.

Differences were significant for stimulus-locked data (in a window of 0–4 s with t = 0 at stimulus onset, 4–181 Hz), revealing two clusters of activity associated with painful heat vs non-painful cold stimulation (Fig. 4a). We observed a positive cluster ($p = .004$), including frequencies from 9.5 to 181 Hz in a time frame of 2.05–4 s, indicating an increase of EEG power in beta-to-gamma (>12 Hz) in painful heat

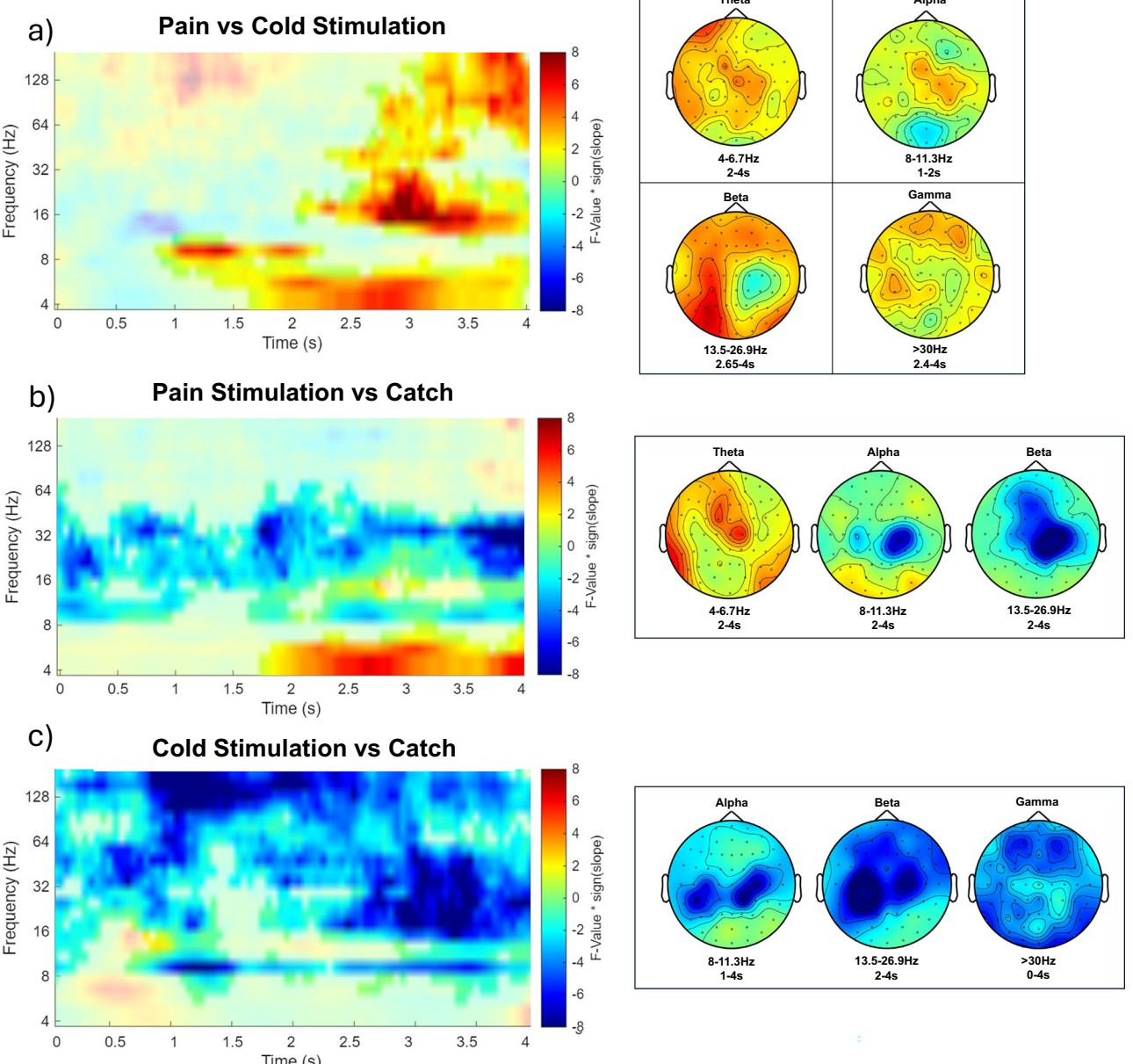

**Fig. 4 | EEG patterns of painful heat and non-painful cold stimulation.** Time-frequency representation represents statistical F-values derived from a two-sided cluster-based permutation test, based on F-statistics ($n = 35$) at each sample (time x frequency), averaged across all EEG channels, with t = 0 at stimulation onset. Panels depict **a** painful heat vs. non-painful cold, **b** painful heat vs. catch, and **c** non-painful cold vs. catch. Pain vs. catch revealed canonical pain-related EEG patterns with theta synchronization and alpha-to-beta desynchronization, whereas cold vs. catch showed desynchronization across alpha-to-gamma frequencies. The direct pain vs. cold contrast revealed higher theta–alpha and beta–gamma power for painful heat. In (**a**), hot colors indicate higher power for painful heat compared to non-painful cold, and cold colors indicate lower power. In (**b**, **c**), hot colors represent power increases with stimulation relative to catch, and cold colors represent decreases. Significant clusters are highlighted, and topographies show F-values at each electrode within significant time–frequency windows.

stimulation vs non-painful cold stimulation. The highest F-value from the repeated-measures ANOVA was F(1,34) = 31.82 ($p < .001$). This sample was observed after 3 s at 16 Hz and had a maximum at channel CP3. In the theta-to-alpha band (4–12 Hz), we found another significant positive cluster of activity ($p = .034$), showing higher EEG power with painful heat vs non-painful cold stimulation. This cluster included frequencies from 4 to 9.5 Hz in a time frame of 0.85–4 s. The highest parametric F-value from the repeated-measures ANOVA was F(1,34) = 23.19 ($p < .001$). This sample was observed after 1.4 s at 8 Hz and had a maximum at channel FCz.

Secondly, we tested stimulation of (painful) heat and (non-painful) cold, separately, against respective catch trial periods (with the

same modality and timing cues) without stimulation. To avoid potential confounds with prediction errors, only correctly cued trials were included in the analysis of stimulation effects on EEG.

For painful heat stimulation versus catch trials (Fig. 4b), differences were significant for stimulus-locked data (in a window of 0–4 s with t = 0 at stimulus onset, 4–181 Hz), revealing two clusters of activity associated with painful heat stimulation. We observed a positive cluster ($p = .047$), including frequencies from 4 to 6.8 Hz in a time frame of 1.75–4 s, indicating an increase of EEG power in theta (4–8 Hz) in painful stimulation. The highest F-value from the repeated-measures ANOVA was F(1,34) = 24.33 ($p < .001$). This sample was observed after 2.65 s at 4 Hz and had a maximum at channel C2. In the alpha-to-beta

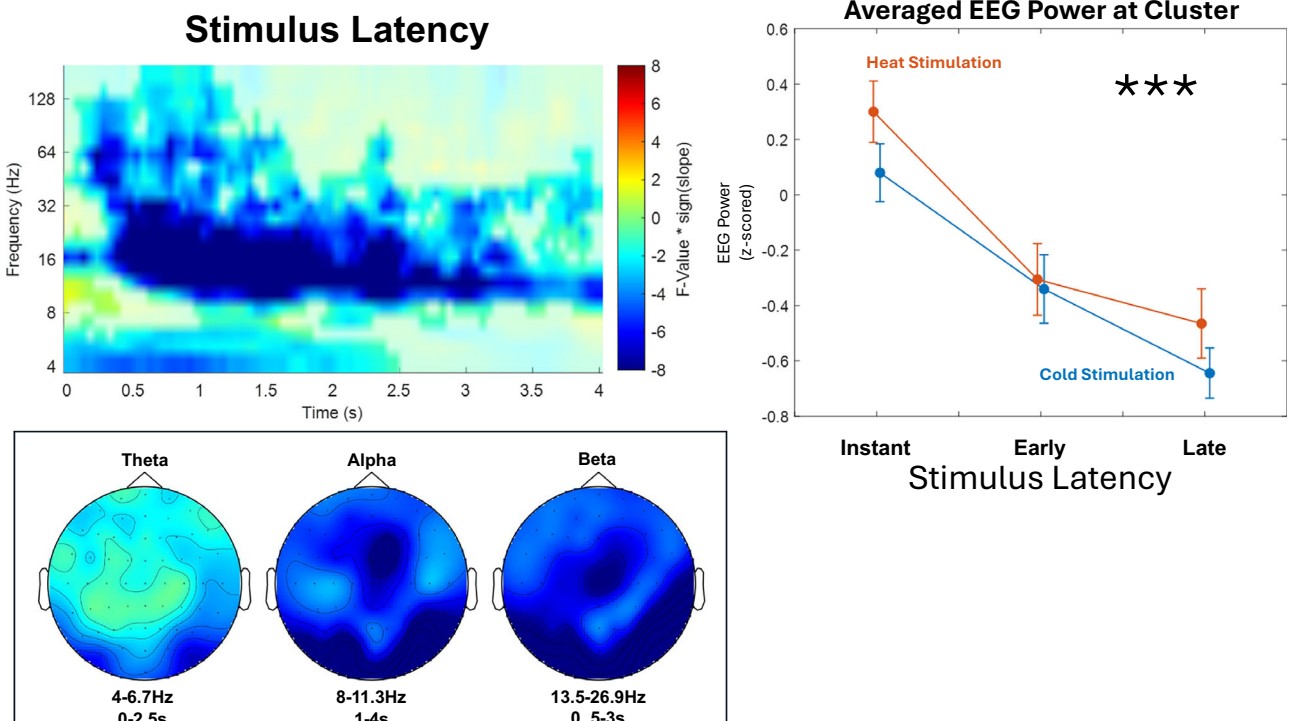

**Fig. 5 | Stimulus latency.** The contrast of actual stimulus latency at stimulation shows desynchronization of alpha-to-beta power predominately at posterior areas. Time-frequency representation represents statistical F-values derived from a two-sided cluster-based permutation test, based on F-statistics ($n = 35$) at each sample (time x frequency) averaged over all EEG channels with t = 0 at the onset of the (painful) heat or (non-painful) cold stimulus. Cold colors represent a negative slope of EEG activity (i.e., a lower power was detected with increased cue-stimulus delays), whereas warm colors represent a positive slope (i.e., a higher power with increased cue-stimulus delays). EEG power was averaged within subjects before group-level statistics. Significant clusters are highlighted. Line graphs represent averaged z-scored EEG power (over all significant samples included in this cluster) for instant, early and late stimulation conditions. Error bars represent SEM across subjects. Topographies represent F-values at each electrode within respective frequency bands in significant time frames. Asterisks indicate significant cluster-corrected main effects of latency (*** indicates $p \leq .001$). Statistical inference is based on cluster-based permutation tests across subjects; line plots are shown for visualization only.

band (8–30 Hz), we found a significant negative cluster of activity ($p < .001$), showing desynchronization of EEG power with pain stimulation. This cluster included frequencies from 8 to 63 Hz in a time frame of 0–4 s. The highest parametric F-value from the repeated-measures ANOVA was $F(1,34) = 45.97$ ($p < .001$). This sample was observed after 4 s at 22.6 Hz and had a maximum at channel Fz.

For non-painful cold stimulation (Fig. 4c), we observed one cluster of activity ($p < .001$), showing desynchronization in the theta, alpha-to-beta, and gamma bands from 4.8 to 180 Hz. The highest parametric F-value from the repeated-measures ANOVA was $F(1,34) = 49.18$ ($p < .001$). This sample was found at 0.85 s and 19 Hz and had a maximum at channel C4.

Painful stimulation is typically associated with increases in the theta band (4–8 Hz), decreases in the alpha and beta band (8–30 Hz), and increases in the gamma band (>30 Hz) activity[45]. When contrasting painful heat with non-painful cold stimulation, we observed widespread increases in EEG power in the theta (4–8 Hz), alpha-to-beta (8–30 Hz), and gamma range (>30 Hz), but no significant alpha-to-beta (8–30 Hz) decreases for pain in the direct modality comparison. By contrast, when each stimulation type was compared against its respective catch condition, both painful heat and non-painful cold stimulation elicited significant alpha-to-beta (8–30 Hz) desynchronization, consistent with canonical somatosensory responses. For gamma activity (>30 Hz), painful heat stimulation did not show a significant increase relative to catch trials. However, in the direct pain vs. cold comparison, gamma power was significantly higher for pain, an effect that may be partly driven by reduced gamma activity during cold stimulation in the present data.

## Stimulus latency

In this experiment, participants were exposed to either (painful) heat or (non-painful) cold stimuli, delivered at varying delays following a cue: instant (0 s), early (2 s), or late (4 s).

As a first step, we tested for the effects of actual stimulus latency (i.e., the duration between cue offset and stimulus onset) on EEG time-frequency data. We considered EEG data from stimulus onset (time-locked to 0 s) to stimulus-offset (time-locked to 4 s) and conducted F-tests contrasting instant, early, and late stimulation and testing its interaction with stimulus modality (heat vs cold).

For the main effect of stimulus latency (Fig. 5), a cluster-corrected dependent samples F-test revealed significant changes in stimulus-locked EEG power (0–4 s at stimulation onset, 4–181 Hz). Data shows a negative cluster ($p < .001$) predominantly in the alpha and beta bands (8–30 Hz), with significant samples ranging from 0 to 4 s, including frequencies from 4 to 181 Hz, exhibiting a negative association between longer stimulus latencies and EEG power during stimulation. The highest parametric F-value from the repeated-measures ANOVA was $F(1,34) = 85.89$ ($p < .001$). This sample was found at 0.95 s at 16 Hz and had a maximum at channel PO8. A topography of statistical F-values indicates that the stimulus latency main effect originates from posterior regions. We found no significant cluster of activity associated with an interaction between stimulus latency and stimulus modality (all $p > .05$).

This represents actual temporal stimulus properties—i.e., the stimulus delay—, which appear to be predominantly encoded at posterior regions in lower frequency bands (theta and alpha-to-beta).

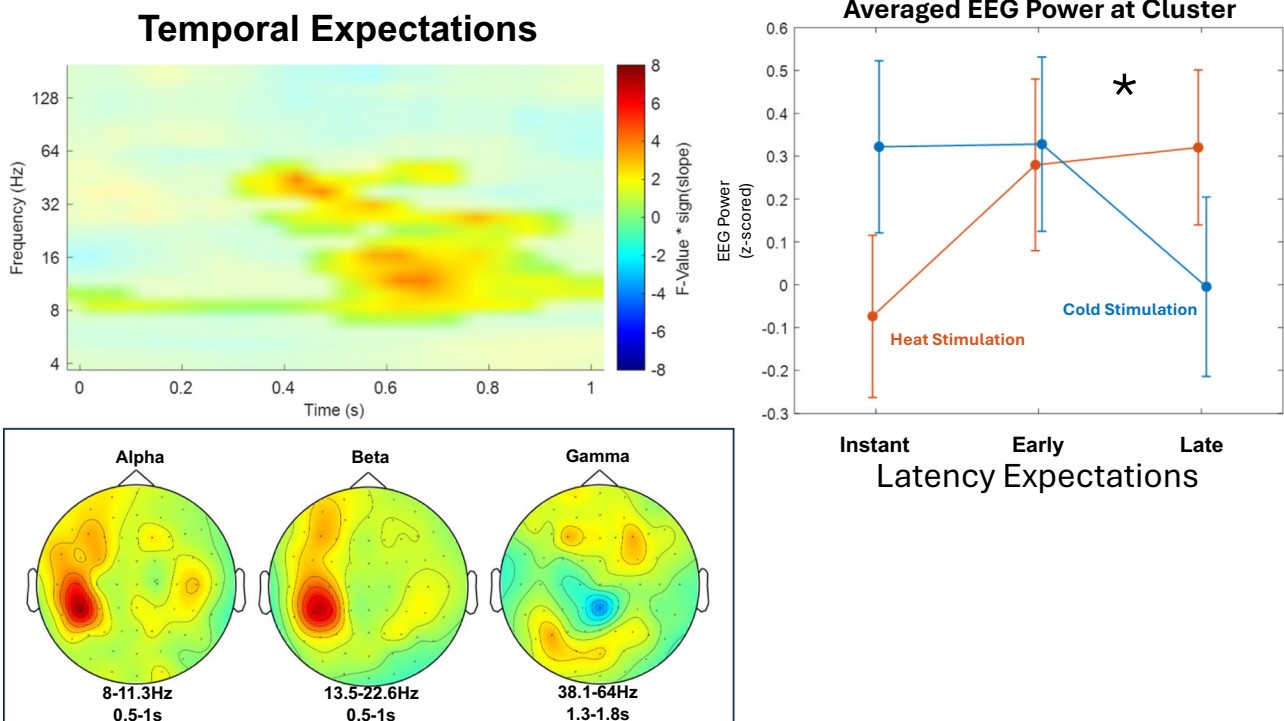

**Fig. 6 | Temporal expectations.** The significant crossed interaction of expected stimulus latency representing temporal expectations and stimulus modality (heat vs cold) shows a synchronization of alpha-to-beta frequencies during cue presentation for (painful) heat and a desynchronization of alpha-to-beta frequencies for (non-painful) cold, representing a crossed interaction. Time-frequency representation represents statistical F-values derived from a two-sided cluster-based permutation test, based on F-statistics ($n = 35$) at each sample (time x frequency) averaged over all EEG channels, with t = 0 at the onset of the probabilistic cue inducing expectations regarding the cue-stimulus delay. Warm colors represent a relative increase of EEG activity for heat vs cold with increasing temporal latency expectations, whereas cold colors represent a relative decrease for heat vs cold stimulation. EEG power was averaged within subjects before group-level statistics. Significant clusters are highlighted. Line graphs represent averaged z-scored EEG power (over all significant samples included in this cluster) for instant, early and late latency expectation conditions. Error bars represent SEM across subjects. Topographies represent F-values at each electrode within respective frequency bands in significant time frames. Asterisks indicate significant cluster-corrected interaction effects of stimulus modality and expectation (* indicates $p \leq .05$; exact $p = .048$). Statistical inference is based on cluster-based permutation tests across subjects; line plots are shown for visualization only.

## Temporal expectations

As a next step, we evaluated how temporal expectations modulated EEG time-frequency patterns. In this study, a cue presented before stimulus presentation conveyed information (probabilistically) whether the next stimulus will be initiated instantly (0 s), early (2 s), or late (4 s), and thus provided information to induce temporal expectations regarding the subsequent stimulus. As this expectation signal might be present during both the cue presentation as well as during stimulus presentation, we conducted cluster-corrected dependent samples F-test both during cue presentation (0–1 s, 4–181 Hz; time-locked to the onset of the predictive cue) and during stimulation (0–4 s, 4–181 Hz; time-locked to the onset of the thermal stimulation).

In the time frame of the presentation of the cue indicating instant, early or late stimulation, dependent samples F-tests (0–1 s, 4–181 Hz; time-locked to the onset of the predictive cue) revealed no significant main effect of expected latency (all $p > .05$). We observed a significant crossed modality x expected latency interaction in the alpha-to-beta bands (8–30 Hz; $p = .048$) with significant samples occurring from 0 to 1 s after cue onset at frequencies between 6.7 and 53.8 Hz, showing a positive association between expected longer latencies and EEG power during cue presentation for (painful) heat stimuli and a negative association between longer expected latencies and EEG power during cue presentation for (non-painful) cold stimuli (Fig. 6). The highest parametric F-value from the repeated-measures ANOVA was $F(1,34) = 21.42$ ($p < .001$). This sample was found at 0.75 s after cue onset and 16 Hz and had a maximum at channel CP3.

The analysis of stimulus-locked data revealed no association between latency expectations and EEG data during stimulation (all $p > .05$). Also, our data revealed no significant interaction between stimulus modality and expected latency in stimulus-locked data (all $p > .05$).

The data presented here provides evidence that temporal expectations regarding subsequent temporal stimulus properties are encoded during cue presentation, whereas EEG activity during stimulus presentation was not modulated by latency expectations.

## Temporal prediction errors

One aim of this study was to investigate whether discrepancies between temporal expectations and actual stimulus latencies would lead to measurable changes in time-frequency EEG data. Specifically, we examined absolute prediction errors, coding a late stimulus with instant latency expectations (a high prediction error) in the same way as an instant stimulus with late latency expectations (also a high prediction error), whereas a stimulus with instant latency expectations would not create a prediction error in instant expectation conditions (see Supplementary Fig. 4 for results with a prediction error coded based on the actual probability at stimulation; see Supplementary Fig. 5 for EEG analysis results with all datasets with 10 or more trials in each PE condition; see Supplementary Fig. 6 for additional time-frequency patterns of absolute prediction errors).

For the main effect of (absolute) temporal prediction errors (Fig. 7), we found an association between stimulus-locked EEG data

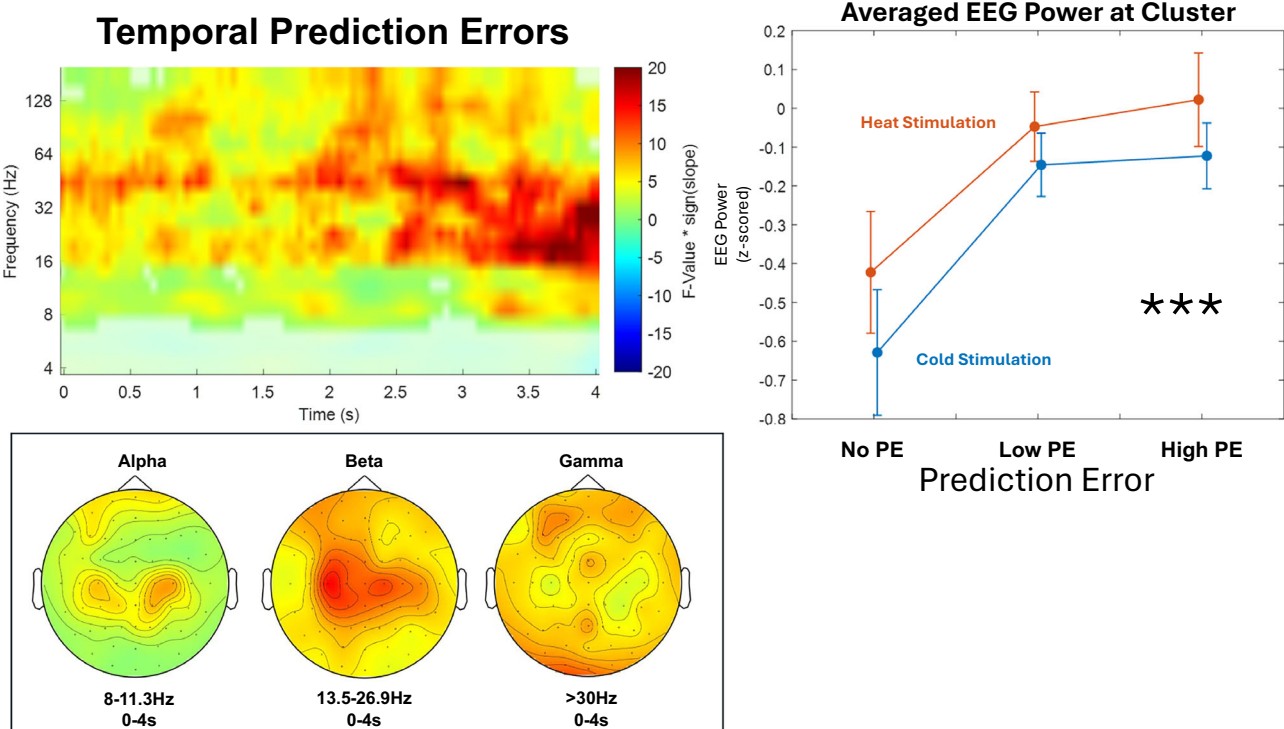

**Fig. 7 | Temporal prediction errors.** The contrast of absolute prediction errors (i.e., the absolute difference between expected and actual cue-stimulus delay) shows a synchronization of beta-to-gamma frequencies during stimulation presentation. Time-frequency representation with statistical F-values derived from a two-sided cluster-based permutation test, based on F-statistics (*n* = 35) at each sample (time x frequency) averaged over all EEG channels, with t = 0 at the onset of the (painful) heat or (non-painful) cold stimulus. Warm colors represent a positive slope of EEG activity (i.e., a higher power was detected with increased absolute prediction errors), whereas cold colors represent a negative slope (i.e., a lower power with increased absolute prediction errors). EEG power was averaged within subjects before group-level statistics. Significant clusters are highlighted. Line graphs represent averaged z-scored EEG power (over all significant samples included in this cluster) for no, low, and high prediction error conditions. Error bars represent SEM across subjects. Topographies represent F-values at each electrode within respective frequency bands in significant time frames. Asterisks indicate significant cluster-corrected main effects of temporal prediction errors (*** indicates *p* ≤ .001). Statistical inference is based on cluster-based permutation tests across subjects; line plots are shown for visualization only.

(0–4 s at stimulation onset, 4–181 Hz) with absolute temporal prediction errors in beta-to-gamma frequencies (>12 Hz) predominantly in the second half of stimulation. This positive cluster (*p* < .001) includes samples ranging from 0 to 4 s and frequencies from >6.7 Hz. The highest parametric F-value from the repeated-measures ANOVA was F(1,34) = 94.13 (*p* < .001). This sample was observed at 4 s and 16 Hz and had a maximum at channel C3. We found no significant interaction between absolute prediction errors and stimulus modality (all *p* > .05).

It is typically theorized that prediction errors are encoded in gamma frequencies (>30 Hz; Bastos et al.). The data presented here demonstrate that temporal prediction errors are encoded predominantly in beta-to-gamma frequencies (>12 Hz) for both (painful) heat and (non-painful) cold stimulation. This implies that a mismatch between expected latencies and actual stimulus latencies leads to increases in beta-to-gamma activity.

## Discussion

The present study investigated temporal expectations and resulting prediction errors, their spectral orchestration, and its effects on pain perception. Additionally, the influence of temporal expectations was investigated, and specifically, whether a dread effect could be associated with actual or expected stimulus delay. One would generally expect prediction errors to influence pain perception. Specifically, unexpected sensory input—that is, prediction errors—have been shown to modulate both the neural processing and the subjective experience of pain[7,12,46]. One possible link between dread and prediction errors is that the aversive anticipation underlying dread may itself generate prediction errors when expected and actual stimulus timings diverge. These errors could amplify negative affect and thus enhance the subjective experience of pain anticipation. However, our study does not provide direct evidence for this mechanism.

In this study, painful heat and non-painful cold stimuli were solely modulated by latency expectations: a higher expected cue-stimulus delay was associated with higher intensity ratings. In contrast, neither actual cue-stimulus delays nor prediction errors had an effect on intensity ratings. This was surprising as a modulation by actual cue-stimulus delays would have been expected by empirical evidence showing higher pain ratings with increased anticipation periods[38,39] and (intensity) prediction errors[7]. The dread phenomenon, where pain is chosen to be experienced rather earlier than later[36,37], can thus—at least in part—be explained via temporal expectation effects leading to an increase in pain with higher expected durations.

To our knowledge, so far, no study has probabilistically manipulated temporal expectations regarding the upcoming stimulus delay. In most paradigms without explicit cues, expectations emerge gradually during the anticipation period, with longer waiting times leading to a growing sense that stimulation must occur soon (e.g., after 5 s: "I have waited a long time now and the pain must come any moment now"). Importantly, the absence of explicit cues does not imply the absence of expectations. Expectations can be formed by elapsed time itself. In contrast, in our paradigm, temporal information is provided upfront by the cue, such that expectations are formed immediately ("I will have to wait 5 s"). While expectations may still evolve as the trial progresses (see Fig. 1, which shows "total probability" and "probability

at stimulation"), providing cue information at trial onset eliminates the confound between expected waiting times and actual waiting times.

The unpredictable condition presented in the study by Clark et al.[38] illustrates the difficulty of disentangling these processes: longer anticipation periods were inherently accompanied by a greater buildup of temporal expectations, such that effects of actual waiting time could not be separated from the gradual generation of expectations during anticipation. Notably, though, Clark et al.[38], also reported no significant difference between shorter anticipation intervals (3 vs. 6 s), which is consistent with our finding of no modulation by actual delays of 0–4 s. This suggests that the classic anticipation–pain association may require longer intervals than those tested here, and we therefore cannot exclude that stronger effects of actual waiting time would emerge with extended delays.

Clark et al.[38] reported P300 modulations by unpredictability without corresponding changes in pain ratings. In our data, we observed no behavioral effect of temporal prediction errors, but prediction errors were represented in beta-to-gamma frequencies (>12 Hz). This suggests that while subjective pain experience is driven by expectation-based modulations, neural changes based on unpredictability and prediction errors reflect the brain's sensitivity to violations of temporal predictions, even when such violations do not translate into altered pain ratings.

Hauck et al.[39], reported that longer cue–pain delays increased pain ratings. However, in their paradigm, longer delays were also associated with higher probabilities of receiving a painful stimulus, effectively confounding waiting time with increased prior confidence about pain. From a Bayesian perspective, this design may have amplified pain ratings through stronger priors rather than through elapsed time alone. By contrast, our cue-based manipulation separates expected from actual delays and controls pain probabilities—even as the trial progresses—by the implementation of catch trials without stimulation, allowing us to demonstrate that it is the expectation of a longer waiting period—not the physical delay itself—that drives the observed increase in pain.

Furthermore, we found an increase in the intensity of the cold percept, which was clearly non-painful, which is also indicative of a top-down temporal expectation process associated with thermal stimulation driving an increased intensity of the percept, instead of, for example, dread for pain. In this study, EEG power modulation by expected latency occurs during cue presentation, but not during stimulation. Within a Bayesian integration model, this suggests that latency expectations influence top-down priors rather than bottom-up sensory processing.

Our data revealed that the temporal and spectral orchestration of predictive timing (prediction, evidence, and prediction errors) is represented in distinct changes of EEG power. Effects of temporal expectations were observed after the predictive cue in the alpha-to-beta band (8–30 Hz), where higher expected latencies led to a crossed interaction: increased alpha-to-beta power during heat stimulation but decreased alpha-to-beta power during cold stimulation. Prediction errors were represented in beta-to-gamma frequencies (>12 Hz). The sensory evidence (i.e., actual stimulus latency) was represented in alpha-to-beta (8–30 Hz) frequencies during stimulation.

Importantly, our EEG analysis revealed divergence with canonical EEG patterns of pain as summarized by Ploner et al.[45]. In line with prior work, the pain vs. catch contrast reproduced the expected profile of theta (4–8 Hz) synchronization and alpha-to-beta (8–30 Hz) desynchronization during painful stimulation but did not show increases in the gamma frequency band (>30 Hz). However, when directly contrasting painful heat with non-painful cold, we observed a different pattern: painful heat was associated with higher theta–alpha and beta–gamma power. Notably, this relative increase seems to be partly driven by decreases in cold stimulation, where we observed desynchronization across alpha-to-gamma frequencies. Thus, while the pain vs. catch analysis captures canonical pain-related oscillatory

responses, the pain vs. cold comparison highlights modality-specific differences that extend beyond this canonical profile.

Previous EEG pain studies have consistently shown that cue-induced expectation effects do not modulate EEG power during pain stimulation, even when behavioral modulations via cue-induced expectations are observed[12,18,29–31]. Instead, pain-related expectations are inconsistently associated with an increase or decrease of lower frequency power (<30 Hz) with increased pain expectations during cue presentation[12,18,29–31]. However, recently Bott et al.[47], could show that cue-induced expectations systematically modulate interregional connectivity during pain stimulation.

In this study, temporal expectations regarding an upcoming pain stimulus modulate the alpha-to-beta signal during cue presentation. In another recent study, agency led to a crossed interaction and to an increase or decrease of alpha-to-beta power depending on pain (treatment) expectations in self-treatment or external-treatment conditions. This indicates that the alpha-to-beta signal appears to involve complex contextual/cognitive factors in the preparation for pain signals.

Therefore, future studies must consider the precise control of these factors to better understand this signal. Here, we demonstrate that this signal is also modulated by variations of temporal expectations, and moreover, differentially encodes for painful heat and non-painful cold stimulation. This difference in sign (i.e., increase in signal with higher expected delays for painful heat/decrease in signal with higher expected delays for non-painful cold) could suggest that the brain differentially processes painful versus non-painful stimuli, reflecting distinct affective or sensory components, especially in the light of threat and dread.

In natural contexts, individuals are unlikely to encounter pain in such a predictable manner - for example, the closest analogy might be the expectation of a needle prick when counting to three before an injection, when the actual prick occurs unexpectedly at two. Despite this artificial setting, the tightly controlled nature of our experiment allowed us to make significant contributions to understanding the basic mechanisms of temporal expectations.

We demonstrated that stimulus ratings were influenced by expected delays rather than actual delay periods. This was observed for both non-painful cold and painful heat stimuli.

A limitation of this study is the categorical treatment of thermal stimuli as either painful heat or non-painful cold. While useful for analysis, this simplification may overlook the continuous nature of thermosensation. Future work should consider continuous intensity scales or include intermediate conditions to better capture the spectrum of thermal perception. Another potential limitation is the maximum cue-stimulus interval of 4 s, which might not be long enough to fully engage prolonged affective anticipation associated with dread. However, the presence of a robust cue effect suggests that participants were sensitive to temporal expectations within this timeframe. A further limitation of the present study is its exclusive focus on signal power, potentially missing complementary insights from phase-based connectivity EEG metrics[47]. Another limitation is that the exploratory Bayesian model assumes static prior means and precision across trials. Due to the block design, where probabilities remain constant, this assumption of static expectations is likely valid. However, in a different approach, one could update prior and likelihood values dynamically based on sensory input and prediction errors, and model these using dynamic models, e.g., Kalman filter[17,48].

## Methods

### Subjects

35 healthy participants were enrolled (mean age 27.9, range 19–40 years, sex: 18 female/17 male). Note that no formal sex- or gender-specific analyses were conducted, as the study was not powered to detect reliable sex- or gender-dependent effects. All participants gave

informed consent and were paid as compensation for their participation. Applicants were excluded if one of the following exclusion criteria applied: neurological, psychiatric, dermatological diseases, pain conditions, current medication, or substance abuse. Participants were recruited via convenience sampling using advertisements on a local job platform, targeting the local community and university population. Potential self-selection effects are inherent to this recruitment strategy. The study was approved by the Ethics Board of the Hamburg Medical Association.

The sample size was determined through a priori power analysis using G*Power (version 3.1.9.4). We informed the power analysis by behavioral effects observed in Strube et al.[30], where expectations were manipulated under similar experimental conditions in the same laboratory. That study reported large effect sizes (Cohen's f > 0.5) for both intensity expectation effects and modulations of pain ratings by agency. Assuming a low correlation among repeated measures (r = 0.2), an alpha level of 0.05, and a desired power of 0.80, the analysis indicated that a sample size of 12 subjects would be sufficient to detect similar effects. However, considering potential differences in expectation modulation—specifically, temporal versus intensity expectations—we increased the planned sample size to 35 subjects. Accounting for an anticipated dropout of four subjects, a final sample size of 31 provides adequate power to detect medium effect sizes (f = 0.3) in our within-subject design.

### Stimuli and task

Thermal stimulation was performed using a $30 \times 30$ mm² Peltier thermode (CHEPS Pathway, Medoc) at two different intensities: (non-painful) cold (20.5 °C) and (painful) heat (46.5 °C) at the left radial forearm. The baseline temperature was set at 30 °C and the rise rate to 8 °C/s. After each block, the stimulated skin area was changed to avoid sensitization. To ensure a clear perceptual contrast between painful and non-painful stimulation, we used cold stimuli (20.5 °C). Pilot testing indicated that warm non-painful stimuli near the neutral threshold (32–34 °C) were perceived inconsistently across pilot participants and exhibited strong habituation effects. In contrast, cold stimuli were reliably perceived as non-painful but salient.

Each trial started with the presentation of a temperature cue (indicating correctly if the next stimulus is a heat or cold stimulus) by the display of "heat" or "cold" on the center of the screen for 1 s. Afterwards, a fractal was presented for 1 s as a temporal expectation cue and probabilistically associated with instant, early, or late stimulation (see Fig. 1a for a contingency table). In test trials, instant, early, and late stimulation (either heat or cold stimulation) was initiated 0, 2, or 4 s after the offset of the temporal expectation cue, respectively. In catch trials, after an anticipation period of 8 s (i.e., the whole duration of a late stimulation trial including anticipation and stimulation), participants were asked to report the information of both cues (temperature and temporal cue). The intertrial interval was set to 2 s.

Trials were presented in 5 blocks (including one initial training block). At the beginning of each block, the three different fractals associated with instant, early, and late stimulation were shown (fractals were randomized for each participant), and participants were asked to learn these associations. Participants were given time until they were sure that they learned the associations. Participants were informed that the temporal cue was correctly predicting the latency until the onset of the pain in most cases, but not in all cases. At the beginning of each block, we asked participants to rate the salience of 2 heat and 2 cold stimuli (4 s stimulation, 46.5 °C for heat and 20.5 °C for cold) on a visual analog scale (VAS) from 0 to 100, where 0 was labeled as "no stimulation" and 100 was rated as "very salient".

During the training block, participants only experienced congruent trials and catch trials in a fixed order, e.g., an instant temporal expectation cue was followed by instant stimulation. The fixed order was as follows: Instant Cold, Instant Cold Catch, Instant Heat, Instant

Heat Catch, Early Cold, Early Cold Catch, Early Pain, Early Pain Catch, Late Cold, Late Cold Catch, Late Pain, Late Pain Catch.

During 4 experimental blocks, trials were presented in a randomized order in two micro blocks. Each micro block contained 4 congruent trials of each congruent cue-stimulus combination (e.g., early cold cue leads to actual early cold stimulation), 1 incongruent trial of each incongruent cue-stimulus combination (e.g., early heat cue leads to instant heat stimulation), and 2 catch trials for each potential cue-stimulus combination. For randomization, we used a sampling method to guarantee that there were no two consecutive trials of the same condition within one block. Catch trials were implemented to ensure that late latency stimulation (4 s) was still improbable in instant and early expectation trials (i.e., probability of stimulation was at 33%) during the progression of a trial, even though stimulation did not occur after 0 and 2 s in instant and early (0 and 2 s) expectation conditions. For example, if stimulation did not occur instantly after 0 s, this changes the probabilities of stimulation at later time points (2 or 4 s) as the trial temporally progresses.

Participants were allowed to take a break between each block without a time limitation. The whole experimental procedure (incl. EEG preparation, informed consent, etc.) had a duration of approximately 3.5–4 h. The participant, the experimenter, and, in some sessions, a trained research assistant were present in the testing room. The experimenter followed a standardized experimental protocol and did not provide any feedback regarding experimental conditions or hypotheses during data collection. Blinding of the experimenter to stimulus modality and timing conditions was not feasible due to the nature of the stimulation procedure; however, all task instructions, stimulus delivery, and data acquisition were fully automated, minimizing potential experimenter-related bias.

### EEG recording

EEG data were acquired using a 64-channel Ag/AgCl active electrode system (ActiCap64; Brain Products GmbH, Germany) placed according to the extended 10–20 system[49]. The EEG was sampled at 500 Hz, referenced at FCz, and grounded at Iz. For removal of ocular movement artifacts, the horizontal and vertical bipolar electrooculogram (EOG) were recorded using the 4 remaining electrodes.

### EEG preprocessing

The data analysis was performed using the Fieldtrip toolbox for EEG/MEG analysis[50]. For preprocessing, data were epoched and time-locked to the onset of the cue signaling whether the upcoming stimulus was a (non-painful) cold or (painful) heat stimulus. Each epoch was centered (subtraction of the temporal mean) and included a time range of 3 s before and 12 s after trigger onset. In 7 subjects, channel P3 was interpolated using the average of the adjacent electrodes, as this channel was found to be affected by very low signal quality in these subjects. The data were re-referenced to a common average of all EEG channels, and the previous reference channel FCz was reused as a data channel.

EEG preprocessing steps were replicated from a previous study from this laboratory to ensure consistency between analyses[30]. We employed a preprocessing approach by Hipp et al.[51], by splitting the data into 2 band-pass filtered sub-sets from 4 to 34 Hz for low frequencies and from 16 to 250 Hz for high frequencies. This enabled efficient separation of low- and high frequency artifacts in subsequent ICA analysis. EEG epochs were visually inspected, and trials contaminated by artifacts due to gross movements or large technical artifacts were removed. Trials contaminated by eye-blinks, heartbeat, muscle activity, technical artifacts, or movements were corrected using an independent component analysis (ICA) algorithm[52–54] after careful inspection of topographies, power spectra, and relation of ICA time courses to the temporal structure of the experiment. Artifactual components were removed before the remaining components were

back-projected and resulted in corrected data. Finally, epochs were visually screened and trials with remaining artifacts were excluded from analysis (see Supplementary Table 2 and Supplementary Table 3 for the number of trials included for each subject, for painful heat and non-painful cold conditions, respectively; see Supplementary Table 4 and Supplementary Table 5 for the number of trials included for each subject, for each trial combination representing stimulus latency (instant, early late), expected stimulus latency (instant, early, late) and PE (no, low, high) conditions for painful heat and non-painful cold, respectively).

## EEG spectral analysis

Spectral analysis was adapted from Hipp et al.[51], and replicates previous analysis from our laboratory[30]. This approach ensured homogenous sampling and smoothing in time and frequency space. We calculated spectral estimates for 23 logarithmically scaled frequencies ranging from 4 to 181 Hz (0.25 octave increments) for the whole epoch. Using the multitaper (DPSS) approach, we set the temporal and spectral smoothing to match 250 ms and 3/4 octave, respectively. For frequencies below 16 Hz, we employed 250 ms temporal windows and varied the number of Slepian tapers to approximate a 3/4 octave spectrum smoothing. We computed the frequency transform using high- and low-frequency data for frequencies above and below 25 Hz, respectively. Analysis was then continued with the combined spectral data after averaging of spectral estimates per block and condition over trials for each subject.

For the baseline correction of time-frequency data, the mean and standard deviation were estimated (for each subject/channel/frequency combination, separately) from −1000 to −500 ms before the onset of the modality cue (indicating if the subsequent stimulus was a cold or heat stimulus). The mean spectral estimate of the baseline was then subtracted from each data point, and the resulting baseline-centered values were divided by the baseline standard deviation[55].

Before further data analysis was performed on the cleaned datasets, the time axis of single trials was shifted to create cue-locked and stimulus-locked data. For cue-locked data, we used the onset of the cue signaling the expected stimulus latency as t = 0. For stimulus-locked data analyses, we used t = 0 at the onset of the heat or cold stimulation, i.e., to the trigger signaling the thermode to initiate a temperature change. To test for a main effect of cold or heat stimulation, we also used EEG data from catch trials, which was locked to the exact time-windows where stimulation would occur in stimulation trials, i.e., 0–4 s after cue offset in instant stimulation trials, 2–6 s after cue offset in early stimulation trials, and 4–8 s after cue offset in late stimulation trials.

## Predictive timing model

Similar to previous fMRI and EEG studies with intensity expectations in pain[12,15], our full model included three experimental within-subject factors (see Fig. 1b) to comprise a simple predictive timing coding model. Here, we adopted this model and translated stimulus intensity, intensity expectations, and prediction errors, and reframed these as stimulus latency, temporal expectations, and temporal prediction errors to represent a simple predictive timing model.

The stimulus latency factor (see Fig. 1b, left column) models the measured response with a simple linear function of the stimulus latency (−1, 0, and 1 for instant, early, and late stimulation, respectively). The temporal expectation factor was defined (see Fig. 1b, center column) linearly from the latency predicted by the cue. Again, conditions with an instant stimulation cue were coded with a − 1, conditions with an early stimulation cue with a 0, and conditions with a late stimulation cue with a 1. The temporal prediction error factor (PE) resulted from the absolute difference of the temporal expectations and the actual stimulus latency (see Fig. 1b, right column).

## Behavioral intensity ratings

Behavioral intensity ratings were analyzed at the single-trial level using linear mixed-effects models (LMEs). Trials were defined by the full combination of stimulus modality (cold vs. heat), predictive cue (instant: 0 s, early: 2 s, or late: 4 s), and actual stimulus latency (instant: 0 s, early: 2 s, or late: 4 s delay), allowing for prediction errors to occur (i.e., the absolute difference between expected and actual cue-stimulus latency). Temporal expectations were modeled as continuous predictors derived from the predictive cues, actual stimulus latency was included as a continuous variable, and absolute prediction error (|expected latency - actual latency|) was entered as an additional predictor. Stimulus modality (heat vs. cold) was included as a categorical effect. Analyses were performed in MATLAB (fitlme function, version 2025a, The MathWorks). LMEs included all predictors simultaneously, with random intercepts and random slopes for each predictor by subject, thus accounting for between-subject variability and accommodating unbalanced trial numbers across conditions. Post hoc comparisons were Bonferroni-corrected for multiple testing. Violin plots were created using the function *daviolinplot* of the MATLAB toolbox DataViz (v3.2.4)[56].

## EEG data analysis

All statistical tests in electrode space were corrected for multiple comparisons using non-parametrical permutation tests of clusters[57].

We wanted to explore positive and negative time-frequency patterns associated with our variations of stimulus latency, temporal expectations, and (absolute) temporal prediction errors, as well as their interactions with stimulus modality, using a repeated-measures ANOVA. A statistical value corresponding to a *p*-value of 0.05 ($F[1,34] = 4.13$) obtained from the repeated-measures ANOVA F-statistics of the respective main effect was used for clustering. Samples exceeding the threshold of $F[1,34] = 4.13$ were clustered in connected sets on the basis of temporal (i.e., adjacent time points), spatial (i.e., neighboring electrodes), and spectral adjacency. Further, clustering was restricted in a way that only samples were included in a cluster which had at least one significant neighbor in electrode space; that is, at least one neighboring channel also had to exceed the threshold for a sample to be included in the cluster. Neighbors were defined by a template provided by the Fieldtrip toolbox corresponding to the used EEG montage.

Subsequently, a cluster value was defined as the sum of all statistical values of included samples. Monte Carlo sampling was used to generate 1000 random permutations of the design matrix, and statistical tests were repeated in time–frequency space with the random design matrix. The probability of a cluster from the original design matrix (*p*-value) was calculated by the proportion of random design matrices producing a cluster with a cluster value exceeding the original cluster. This test was applied two-sided for negative and positive clusters, which were differentiated by the average slope of the estimated factors, i.e., an increase in EEG power with factor levels 1, 2, and 3 would be regarded as a positive cluster, whereas a decrease in EEG power with factor levels 1, 2, and 3 would be considered a negative cluster of activity.

The within-subject stimulus latency factor (which was coded as increasing linearly with increasing stimulus latency) was tested stimulus-locked from 0 to 4 s. The within-subject temporal expectation factor, which was coded as increasing linearly with the cued stimulus latency, was tested latency-cue-locked from 0 to 1 s and stimulus-locked from 0 to 4 s. The absolute prediction error was coded as the absolute difference between stimulus latency and temporal expectations (see Fig. 1b for details) and was tested during stimulus presentation. In addition, we tested correctly cued non-painful cold and painful heat stimuli, respectively, against the respective time periods of catch trials, as well as non-painful cold vs painful heat.

## Reporting summary

Further information on research design is available in the Nature Portfolio Reporting Summary linked to this article.

## Data availability

Data for this study are publicly available on https://osf.io/tajch/. Project https://doi.org/10.17605/OSF.IO/TAJCH[58].

## Code availability

Code for this study is publicly available on https://osf.io/tajch/. Project https://doi.org/10.17605/OSF.IO/TAJCH[58].

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

## Acknowledgements

C.B. and A.S. are supported by the Deutsche Forschungsgemeinschaft (DFG) SFB 289 project A02 (project ID 422744262–TRR 289). C.B. is supported by European Research Council (ERC) grant AdG-883892-PainPersist. A.S. is supported by a DFG Walter-Benjamin-Scholarship (project ID 531626241). We thank Marie Habermann for helpful comments on the manuscript.

## Author contributions

Conceptualization, data curation, software, formal analysis, investigation, visualization, methodology, writing—original draft, project administration, writing—review and editing, A.S.; conceptualization, resources, formal analysis, supervision, funding acquisition, validation, visualization, methodology, project administration, writing—review and editing, C.B.

## Funding

## Competing interests

The authors declare no competing interests.
