## [Transparent Peer Review file · Nature Communications]

Temporal Predictions Shape Somatosensory Perception

Corresponding Author: Dr Andreas Strube

Version 0:

Reviewer comments:

Reviewer #1

(Remarks to the Author)

In this paper, the authors examine the impact of temporal delay and temporal delay expectations/prediction errors on pain perception and associated EEG activity. I found this to be an excellent paper. There is limited research on the temporal processing of pain and pain expectation, and this work addresses an important gap. The computational modeling offers a compelling perspective, and the experimental design is both simple and robust. I thoroughly enjoyed reading it!

My comments are mostly minor and could be addressed by refining the introduction and discussion. The extent of rewriting required may vary depending on how much the authors are willing to revise, but this represents an opportunity to clarify and strengthen the overall message.

1. Definition of the "Dread" Effect

The authors describe the dread effect as the increased pain perception associated with longer expected delays. However, traditionally, the dread effect refers to a preference for experiencing pain sooner to avoid the aversiveness of anticipation. It is tied to decision-making processes and the desire to minimize the discomfort of the anticipatory period. While the two phenomena are both related to the timing of painful events, they are conceptually distinct. The paper's interpretation of dread as increased pain perception during longer delays is a labeling issue rather than a major conceptual error. However, certain interpretations in the discussion may overextend the definition of dread. For instance:

"This suggests that modulations in the pain experience (which are typically associated with dread) are associated with somatosensory expectation effects, regardless of a "dreading" delay period. Furthermore, we found an increase in the intensity of the cold percept, which was clearly non-painful, which is also indicative of a top-down temporal expectation process associated with aversive stimulation driving an increased intensity of the percept, instead of the dread for pain."

I disagree with the claim that modulations in pain experience are "typically associated with dread." If the authors are aware of specific references that support this, it would be helpful to include them. Otherwise, it may be more accurate to distinguish the reported effects from dread explicitly and consider introducing a different term. The relationship to dread can still be explored in the discussion.

That said, I found the observed effect of increased pain (and cold) perception for longer delays is fascinating, particularly when viewed through a Bayesian lens. Regardless of the terminology, this phenomenon is compelling and warrants publication.

2. Conceptual Disconnect Between the Two Parts of the Paper

The paper appears to focus on two distinct aspects: (1) the increased perception of pain associated with longer expected delays and (2) prediction errors. These two parts are conceptually difficult to reconcile. Initially, one might expect that increased pain perception could be linked to prediction errors, but the results suggest otherwise. Prediction errors emerge as a separate phenomenon. The relationship between these two components could be clarified with a more cohesive narrative in the discussion.

3. Definition of prediction error

However, I think that there is a more fundamental problem with the way the authors have set their prediction errors as depending on a priori probabilities (right panel of figure 1b). Indeed, right panel of figure 1b is generated from “total probability” in table 1a. But total probability is just the a priori probabilistic design of the experiment. The probabilities that participants experience are the “probability at stimulation”. The inverse of those probabilities are the prediction errors that participants experience.

4. Interaction in Figure 7

The interaction depicted in Figure 7 is puzzling. Why would expectations have opposite effects for heat and cold? This inconsistency is intriguing and merits additional discussion or clarification to help readers understand this unexpected result.

Reviewer #2

(Remarks to the Author)

This study examines how temporal expectations influence pain perception, showing that expected delays increase perceived intensity while actual delays do not. EEG results indicate that expected delays modulate alpha/beta activity during anticipation, while actual delays affect activity during stimulation. These findings underscore the role of alpha/beta oscillations in the top-down modulation of sensory processing. While the study’s focus on temporal expectations in the context of pain perception is new, its relevance within theoretical frameworks such as predictive coding or its implications for clinical pain states are not clear to me yet. Additionally, the explanation of statistical methods lacks sufficient detail and clarity, limiting the transparency and interpretability of the analyses.

Major Concerns

- The significance of the predictive coding (PC) framework in the present study is unclear. While this framework is mentioned extensively throughout the manuscript, its significance seems limited to motivating the different contrasts, i.e., effects of actual delays, expected delays, and prediction errors. However, the presence of these effects alone does not provide evidence for a predictive coding mechanism. This would at least require demonstrating that prediction errors lead to changes of future predictions – an aspect not addressed in the study. Repeated reference to the PC framework is distracting and potentially misleading. For example, the claim “This represents a full predictive coding model, ...” in the discussion is unwarranted and should be reconsidered or omitted.

- The comparison of the expectation precision modulation model and the expectation intensity shift model is problematic for two key reasons: First, I understand that, according to Weber’s law, the absolute precision of time period estimates decreases with longer actual time periods. However, it is unclear how this implies that the precision of pain intensity expectations decreases with longer anticipated waiting periods. Pain intensity expectations are cue-induced, and this cue information is unlikely to “wear off” over short durations, such as those in the present experiment. Second, the methods used for constructing and comparing the expectation intensity shift- and expectation precision modulation models are inadequately explained. Bayes rule can be stated as $p(\theta | D) \propto p(D | \theta)p(\theta)$. Here, the posterior $p(\theta | D)$ is a probability density function (PDF) of model parameters θ given the data, the likelihood $p(D | \theta)$ is the PDF of the data given the model parameters θ , and the prior $p(\theta)$ is a PDF reflecting a priori beliefs about values model parameters can assume. Note that the posterior is proportional to the product of two PDFs of two distinct random variables, D and θ . By contrast, the posterior means and variances specified in Eq. 1-3 result from a mere multiplication of two PDFs of a single random variable. Without further explanation, this appears like a misapplication of Bayes rule. Greater mathematical rigor is needed to clarify how Bayes’ rule was applied, including explicit definitions of model parameters (θ), data (D), and the likelihood model. Given these conceptual and technical shortcomings - and considering the analysis is not central to the paper’s primary conclusions - I suggest removing this section.

- The discussion claims that “previous EEG pain studies have consistently shown that cue-induced expectation effects do not modulate EEG signals during pain stimulation ...”. This is not true. For instance, Bott et al. (2023, Science Advances) demonstrated that cue-induced expectations modulate interregional connectivity during pain stimulation. The study’s exclusive focus on signal power should be noted as a limitation.

- The discussion states that the paradigm “eliminates the confound of expected waiting times and actual waiting times”, but this requires clarification. As I understand it, by presenting cues about the waiting time, expectations about waiting times and thus their confound with actual waiting times is introduced in the first place. However, it is also true that the paradigm allows for the disentanglement of the two factors.

- The description of the repeated measure ANOVA in the results and methods sections is unclear (“One-way interaction” is not standard terminology. What are the ANOVA factors? How was the prediction error effect tested specifically? Why is the data summarized in a $2 \times 35 \times 9$ “matrix”? While expected and actual durations are categorical variables with three levels, the phrase “measured response [is modeled] with a simple linear function of stimulus latency” implies these variables are treated as continuous). Providing the code for the analyses would greatly facilitate their understanding (see comment below).

- To promote transparency and reproducibility, the analysis code and data should be made publicly available and easily accessible.

- A priori sample size calculations should be provided. Note that lower power entails a lower likelihood of positive effects to reflect true effects.

Minor concerns

- Are predictive coding principles truly fundamental to all brain processes, or primarily relevant to perception-related processes? (Lines 43-44)
- Timing may be a critical factor in clinical pain contexts, but it is unclear how this specifically motivates the present study (Lines 61-68).
- Why were non-painful cold stimuli used instead of non-painful warm stimuli?
- The coloring of the Table in Figure 1a (color intensity ~ actual latency) is inconsistent with that in Figure 1b (color intensity ~ expected latency).
- Line 194: The phrase "...modulate sensory perception in a way that it is experienced as more intense" is awkward and suggests unintended links to consciousness research ("experience of perception").
- The bar graphs in Figures 3a and 3b reveal little about the data, and the lower cut-offs seem arbitrary. Consider using box plots or raincloud plots instead.
- Line 293: Replace "time points" with "time intervals"?
- For the assessment of stimulation effects on EEG, why were only correctly cued trials used?

Reviewer #3

(Remarks to the Author)
Key Results

This study operationalises a predictive timing paradigm to examine how actual and expected pain stimulus latency influence subjective pain intensity ratings across hot and 20.5-degree (cold) stimulus modalities. By implementing computational Bayesian modelling, the authors investigate the role of prior expectations and precision in shaping pain perception within a predictive coding framework, particularly in relation to the dread effect - the idea that prolonged anticipation amplifies pain perception. EEG data were used to explore the neural signatures of probabilistic inference processes. The results indicate that actual stimulus latency does not significantly affect ratings, but temporal expectations do, such that delayed stimulation was rated as more painful than instant stimulation. Bayesian model selection supported a prior expectation intensity shift model over a prior expectation precision modulation model, suggesting that the content of top-down predictions exerts a stronger influence on ratings than their certainty. The same pattern was observed for the 20.5 degree ("cold") stimulation suggesting that the effect of temporal expectations is not pain-specific.

The key EEG data showed that temporal expectations modulated alpha-to-beta frequencies at cue presentation, but not at stimulus delivery, supporting the argument that they bias prior expectations rather than the processing of ascending nociceptive signal. Temporal prediction errors were expressed in the beta-to-gamma frequency range.

The key EEG data showed that temporal expectations modulated alpha-to-beta frequencies at cue presentation, with a crossed interaction where activity increased for heat and decreased for cold. No significant modulation of alpha-to-beta power was observed at stimulus delivery, supporting the argument that they bias prior expectations rather than the processing of ascending nociceptive signal. Temporal prediction errors were expressed in the beta-to-gamma frequency range, particularly in the gamma range, during the second half of stimulus presentation.

Validity

The experiment offers a valid test of the hypotheses. The rationale is very clear and followed up well by comprehensive analyses, which closely follows the stated rationale. The behavioural modelling is particularly impressive, and all results are examined thoroughly, and interpreted fairly.

A few minor corrections can help increase the validity:

- The 'cold' stimulus is referred to as 'aversive' and 'unpleasant'. Since 20 degrees is not a low temperature this designation should be removed unless it can be supported by evidence.
- The main effect of stimulus intensity on the signal at stimulus delivery (line 368 details the statistical model) is not reported – was there any difference between 'hot' and 'cold'?
- The maximum cue-stimulus interval of 4 seconds in this study may not engage the prolonged affective anticipation necessary to elicit dread-related modulation of pain perception. The null effect of actual stimulus latency may therefore

reflect an insufficient experimental latency window, a possibility which should be acknowledged as a caveat.

- The modelling assumes that the mean and precision of prior expectations are static across trials of equal expectancy conditions. A more biologically plausible implementation would involve a hierarchical precision estimation process, where precision is continuously updated in response to ongoing sensory evidence and past prediction error rather than being pre-determined by task conditions. The authors may wish to consider the assumption as a potential limitation.

Significance

How the effect of dread can be operationalised within predictive processing mechanisms, and how it is implemented computationally and neurally, is a fascinating topic and one that the authors have done an excellent job addressing. The study contributes to our understanding of predictive coding in pain perception by addressing the topic of temporal predictions and prediction errors which has been relatively neglected.

Data and Methodology

The method and analysis are professionally executed.

Analytical Approach

The analysis was executed professionally. Given the core emphasis on Bayesian modelling and the theoretical significance of some null effects we were curious whether the authors might consider employing Bayesian rather than frequentist statistics.

Suggested Improvements

- The authors should report how many trials were included in each condition/participant
- The behavioural analysis reports interactions with stimulus intensity (hot/cold); it should also unpack/explain this interaction.
- Comparing hot/cold separately against catch trials is helpful, but titling Figure 5 'main effect' could be misleading, because this term is typically reserved to the results of a factorial analysis. That figure could improve by including the main effect from the F test in line 368.

Clarity and context

The manuscript is well-structured and provides a clear account of the experimental approach and findings. However, the categorical treatment of thermos-sensation introduces potential conceptual and statistical ambiguities that are not explicitly acknowledged.

References

The manuscript cites relevant literature on predictive coding, Bayesian inference, and pain perception, but the discussion on precision weighting could be better contextualised within the broader hierarchical inference literature. Specifically, research on error-driven precision updates and adaptive inference mechanisms should be referenced to clarify how the study's approach differs from models that incorporate trial-by-trial learning of precision estimates.

Reviewer #4

(Remarks to the Author)

Version 1:

Reviewer comments:

Reviewer #1

(Remarks to the Author)

I am satisfied with the authors' revisions and have no additional comments.

(Remarks on code availability)

Reviewer #2

(Remarks to the Author)

With their revisions, the authors have substantially improved the manuscript. Below are my remaining comments.

- The revised framing and reduced emphasis on Predictive Coding are now more appropriate.
- The explanation of the Bayesian methodology is clearer, and the provided references are helpful. However, the manuscript would benefit from a more rigorous presentation of the approach, e.g., in the supplementary materials. Specifically, explicitly stating the prior, likelihood, and posterior densities (along the lines of Körding and Wolpert, 2004) together with their respective parameters (means and standard deviations) would aid understanding. Moreover, in my opinion, the following statement is problematic: “, where priors and likelihoods represent internal beliefs, not random variables in a statistical inference framework”. Priors and likelihoods, are, by definition, associated with random variables. It is these random variables which are used in conjunction with Bayes theorem to model internal beliefs and the subjective percept. Lastly, in line 924, the procedure for estimating the parameters should be described more precisely. E.g., explicitly list the parameters which are estimated and state the optimization criterion used.
- Regarding the potential confound of expected and actual waiting times, the authors have added an explanatory paragraph, which makes their point clearer (lines 553 - 559). However, in its current form, the added text appears somewhat unconnected with the rest of the explanation. I suggest explicitly stating that the absence of expectation cues does not imply the absence of expectations, if this is what the authors mean.
- The description of the statistical model has improved. However, should it not be “ratings were averaged within each combination of ” rather than “across all combinations of” (line 807)?
- There is no code available under the provided link. Providing the study's code and would enhance its transparency and reproducibility.

(Remarks on code availability)

Reviewer #3

(Remarks to the Author)

The available evidence from prior research suggests that participants tend to anticipate greater pain when it's expected to occur after a delay—possibly accounting for the so-called “dread effect,” in which people would rather experience pain sooner than later. It's intriguing to consider whether this very expectation might amplify the subjective intensity of pain, and it's worth exploring how expectations operate—whether by shifting the mean or altering the variance. Since the behavioral finding runs counter to intuition, the entire manuscript deserves a close, critical read. After reviewing the revised version, several significant concerns emerged.

Major Comments: Interpretation and Modeling

The mean shift model strikes me as a relatively straightforward application of predictive coding principles to these findings. Given that participants report more intense pain when they anticipate a longer delay—and considering that a prior expectation with a higher mean would naturally lead to a higher posterior—the idea of modeling this as a shift in the mean seems both logical and grounded. It offers a solid theoretical framework, especially valuable if the goal is to test alternative explanations of the data.

That said, I'm increasingly unconvinced that the precision-shift model offers a meaningful comparison. While I understand the premise that priors might become less precise with longer expected delays, it's not clear how this imprecision could lead to a mean-shifted posterior. The authors could strengthen their case by more convincingly explaining why this model represents a plausible mechanism, rather than simply re-describing the ANOVA results through a more complex lens.

The discussion of temporal uncertainty should also be contextualized with prior work, particularly Clark et al. (2008), who demonstrated that elevated pain ratings weren't attributable to temporal uncertainty alone. Additionally, the authors suggest their model supports the idea that “cognitive/affective processes alter the valence of late-expected stimuli,” but given the lack of an interaction between expectation and modality (line 173), I'm unsure why a valence-based explanation is justified.

The EEG findings are informative in showing that expectations influence brain activity at the moment the cue is processed but not when the stimulation is processed. This interesting difference between the effect of expectations in two time points should be backed up by including cue and stimulus in the same analysis. However, the broader mechanism underlying the key behavioral effect remains unclear.

Major Comment: Manipulation Check

The experiment uses a factorial design, with modality as a critical variable. This factor is included in the behavioral analysis and again in analyzing temporal delay (line 412). But oddly, it's missing as an independent variable in the immediately preceding EEG analysis. Given the study design, the appropriate test for whether EEG responses differ by modality would be a direct comparison of heat vs. non-painful stimulation, with that contrast featured prominently in the main text (rather than being relegated to the supplementary material).

More troubling, the manipulation check fails to replicate the findings the authors committed to replicating—namely, the pattern identified in Ploner et al.'s (2017) review: increased theta and decreased alpha-beta. While this pattern does emerge in the heat vs. catch trial comparison, that analysis is confounded by the mere presence versus absence of stimulation. The cleaner test—heat vs. control stimulation—does not replicate this expected result, even though it's the more appropriate comparison for a factorial design involving modality.

Major Comment: Reporting Statistics

- Lines 157, 167, and 814 suggest that just one ANOVA was conducted, but from lines 805–821, it's still unclear exactly which statistical tests were run. Were all main effects entered simultaneously? If so, the design would likely be unbalanced (i.e., unequal numbers of trials across some of the 9 conditions). How was this addressed?
- Including both prediction error and latency in the same analysis seems problematic, since the PE calculation combines different conditions (e.g., high PE for late onset/0s cue and early onset/4s cue). Did the authors run multiple two-factor models?

- Lines 178 and 183 suggest that real and expected delay were entered together, yet the degrees of freedom reported for their interaction with modality are (1,34), not (2,34) as would be expected with 2 modalities and 3 levels of latency. This discrepancy needs clarification.

- Results from normality tests and information on how violations of sphericity were handled should be reported.

Minor Comments

- Line 210 contains a typo: “This is at odds with interpretations explaining increased pain by higher, instead,” – sentence is incomplete or awkwardly phrased.

- Figure 1: For “probability at stimulation,” the catch trials should list “N/A” instead of “100%,” since no stimulation occurs.

- EEG trial counts: Some condition counts are reported as fewer than 10 trials. The authors should reassure readers that each condition included in Figure 8 had a sufficient number of trials for robust analysis—typically more than 10, and ideally over 20, as per common practice (see Supplementary Table 1).

- While the paper rightly emphasizes the novel design element—separating expectation from actual delay via the cue—it would be valuable to relate the EEG results back to prior M/EEG studies that examined actual delay. That comparison would help clarify whether earlier findings were truly driven by expectations or were confounded by the delay itself.

(Remarks on code availability)

Version 2:

Reviewer comments:

Reviewer #3

(Remarks to the Author)

The revised manuscript represents a clear improvement aligned with reviewer feedback, but this increased clarity raises important concerns about the modelling.

On temporal uncertainty, they now reference Clark et al. (2008) and position their results in relation to that literature. They also further contextualise their EEG findings relative to Ploner et al. (2017), clarifying the basis for comparison in that review and discussing why a heat/cold contrast can yield a different profile without contradiction.

The revisions provide clearer language and interpretations of results and a cleaner distinction between actual and expected latency, making the central arguments easier to follow. The revision explains how separating expected from actual delay addresses prior designs that bundled these factors and clarifies why short-delay nulls can arise under such confounding.

The authors have added clarifications to how their precision-modulation account yields a mean shift in the posterior conditional on a prior < likelihood asymmetry, and helpfully further specified the assumptions accountable for its behaviour, making explicit the directional assumptions used in their illustrative figures.

However, the implementation of the “expectation precision modulation model” is still problematic. The model makes a critical assumption that the expected mean pain (the prior mean) is lower than the sensory likelihood (via `p_offset`). The authors do not discuss this in any length, but this assumption is necessary to enable reduced prior precision to shift the posterior upward for longer expected delays. While it is entirely possible that sometimes expected pain would be lower, in other times it may be higher, exceeding input across trials. It is not obvious why Bayesian updating would not centre expectations exactly on the intensity of delivered pain, especially considering that uniform stimulation intensities are used (heat/cold).

The “expectation intensity shift model” directly shifts the prior mean upward with delay (via `p_shift`) to fit observed higher ratings, but provides no cognitive or theoretical basis for why expected latency would systematically inflate intensity expectations. Precision modulation has firm support in Bayesian perception accounts, where longer expected delays diminish prior precision under temporal uncertainty. The mean-shift mechanism lacks similar grounding, which makes it read as an add-on that enables a dread-like effect, but not derived from a specified generative link between delay cues and intensity expectations. Without an explicit rationale that elevates it to a general structural (dread-like) feature in line with the principal computational frameworks, its mechanistic and epistemic basis remains uncertain, and is further challenged by the same monotonic effect in the non-painful cold condition.

These issues seem to stem from an underlying epistemic oversight. Both models incorporate assumptions that presuppose that the dread effect exists in order to illustrate it in the data, rather than deriving a model structure from independent Bayesian principles that naturally reproduces it in line with predictive and Bayesian perception accounts. This arguably makes the modelling tautological, in that it is designed to fit behaviours while assuming the effect, rather than to fit the effect assuming a theory of cognition.

As a result, the modelling appears more like parameterisations describing data structure, rather than explaining why that structure occurs. It is recommended that the authors either 1) reframe the focus of the modelling from explanatory to descriptive, or 2) develop a strong rationale that can justify the assumption of a general prior-mean-shift mechanism in the context of Bayesian perception, and/or the assumption that a general mechanism may deflate pain expectations, moving the prior mean below the likelihood mean, so that the latency-dependent precision reduction may push posteriors upwards.

These constitute plausible challenges to the authors’ interpretation with respect to epistemic parsimony, with implications for hypothesis testing. An acknowledgement of this theoretical difficulty in the main text would hedge against strong critiques of the Bayesian model selection results on theoretical grounds.

(Remarks on code availability)

N/A

Version 3:

Reviewer comments:

Reviewer #3

(Remarks to the Author)

The authors have addressed all of our comments.

(Remarks on code availability)

The authors have addressed all of our comments.

Point-by-point responses to the reviewers:

Reviewer #1:

In this paper, the authors examine the impact of temporal delay and temporal delay expectations/prediction errors on pain perception and associated EEG activity. I found this to be an excellent paper. There is limited research on the temporal processing of pain and pain expectation, and this work addresses an important gap. The computational modeling offers a compelling perspective, and the experimental design is both simple and robust. I thoroughly enjoyed reading it!

My comments are mostly minor and could be addressed by refining the introduction and discussion. The extent of rewriting required may vary depending on how much the authors are willing to revise, but this represents an opportunity to clarify and strengthen the overall message.

1. Definition of the "Dread" Effect

The authors describe the dread effect as the increased pain perception associated with longer expected delays. However, traditionally, the dread effect refers to a preference for experiencing pain sooner to avoid the aversiveness of anticipation. It is tied to decision-making processes and the desire to minimize the discomfort of the anticipatory period. While the two phenomena are both related to the timing of painful events, they are conceptually distinct. The paper's interpretation of dread as increased pain perception during longer delays is a labeling issue rather than a major conceptual error. However, certain interpretations in the discussion may overextend the definition of dread. For instance:

"This suggests that modulations in the pain experience (which are typically associated with dread) are associated with somatosensory expectation effects, regardless of a "dreading" delay period. Furthermore, we found an increase in the intensity of the cold percept, which was clearly non-painful, which is also indicative of a top-down temporal expectation process associated with aversive stimulation driving an increased intensity of the percept, instead of the dread for pain."

I disagree with the claim that modulations in pain experience are "typically associated with dread." If the authors are aware of specific references that support this, it would be helpful to include them. Otherwise, it may be more accurate to distinguish the reported effects from dread explicitly and consider introducing a different term. The relationship to dread can still be explored in the discussion.

That said, I found the observed effect of increased pain (and cold) perception for longer delays is fascinating, particularly when viewed through a Bayesian lens. Regardless of the terminology, this phenomenon is compelling and warrants publication.

We agree with the reviewer that our previous use of the term "dread" was not precise. We now consistently refer to our observed effect as a "temporal expectation effect". We revised the introduction and discussion accordingly to avoid overextension and clarify the observed effects. (See lines 19f., 86ff., 119f., 210ff., 563f.)

2. Conceptual Disconnect Between the Two Parts of the Paper

The paper appears to focus on two distinct aspects: (1) the increased perception of pain associated with longer expected delays and (2) prediction errors. These two parts are conceptually difficult to reconcile. Initially, one might expect that increased pain perception could be linked to prediction errors, but the results suggest otherwise. Prediction errors emerge as a separate phenomenon. The

relationship between these two components could be clarified with a more cohesive narrative in the discussion.

We thank the reviewer for raising this conceptual point and changed the narrative in the discussion accordingly. In the revised discussion, we acknowledge that dread and prediction errors may interact, as aversive anticipation could heighten sensitivity to temporal mismatch. We now explicitly discuss this possibility in the revised manuscript. (See lines 527ff.)

3. Definition of prediction error

However, I think that there is a more fundamental problem with the way the authors have set their prediction errors as depending on a priori probabilities (right panel of figure 1b). Indeed, right panel of figure 1b is generated from “total probability” in table 1a. But total probability is just the a priori probabilistic design of the experiment. The probabilities that participants experience are the “probability at stimulation”. The inverse of those probabilities are the prediction errors that participants experience.

As requested, we have added supplementary analyses using the “probability at stimulation” metric to calculate prediction errors (see Supplementary Analysis 2). As can be seen the results remain almost identical when using the probabilities at stimulation, which is expected as our task design was optimized to keep both probabilities quite similar. Therefore, we would suggest to report prediction errors derived from a priori probabilities in the main results, while presenting the analyses based on “probability at stimulation” in the supplementary material. (See lines 199ff.)

4. Interaction in Figure 7

The interaction depicted in Figure 7 is puzzling. Why would expectations have opposite effects for heat and cold? This inconsistency is intriguing and merits additional discussion or clarification to help readers understand this unexpected result.

We agree that this interaction is intriguing. We now include an expanded discussion of potential mechanisms, including differences in motivational salience of threatening / dreadful stimuli in the discussion. (See lines 599ff., 613ff.)

Reviewer #2:

This study examines how temporal expectations influence pain perception, showing that expected delays increase perceived intensity while actual delays do not. EEG results indicate that expected delays modulate alpha/beta activity during anticipation, while actual delays affect activity during stimulation. These findings underscore the role of alpha/beta oscillations in the top-down modulation of sensory processing. While the study’s focus on temporal expectations in the context of pain perception is new, its relevance within theoretical frameworks such as predictive coding or its implications for clinical pain states are not clear to me yet. Additionally, the explanation of statistical methods lacks sufficient detail and clarity, limiting the transparency and interpretability of the analyses.

Major Concerns

- The significance of the predictive coding (PC) framework in the present study is unclear. While this

framework is mentioned extensively throughout the manuscript, its significance seems limited to motivating the different contrasts, i.e., effects of actual delays, expected delays, and prediction errors. However, the presence of these effects alone does not provide evidence for a predictive coding mechanism. This would at least require demonstrating that prediction errors lead to changes of future predictions – an aspect not addressed in the study. Repeated reference to the PC framework is distracting and potentially misleading. For example, the claim “This represents a full predictive coding model, ...” in the discussion is unwarranted and should be reconsidered or omitted.

We thank the reviewer for pointing out this issue. We have now removed overly strong claims such as “full predictive coding model” and instead emphasize that our model tests behavioral and neural signatures of temporal expectations and prediction errors, which are conceptually aligned with—but not exclusive evidence for—predictive coding. The introduction now references our previous predictive coding studies (e.g., Strube et al., 2021) as a motivation, but does not overextend on PC anymore. We also refrained from overextending on PC in the Results part. (See lines 86ff., 357ff., 489ff., 529, 794)

- The comparison of the expectation precision modulation model and the expectation intensity shift model is problematic for two key reasons: First, I understand that, according to Weber’s law, the absolute precision of time period estimates decreases with longer actual time periods. However, it is unclear how this implies that the precision of pain intensity expectations decreases with longer anticipated waiting periods. Pain intensity expectations are cue-induced, and this cue information is unlikely to “wear off” over short durations, such as those in the present experiment.

Second, the methods used for constructing and comparing the expectation intensity shift- and expectation precision modulation models are inadequately explained. Bayes rule can be stated as $p(\theta / D) \propto p(D / \theta)p(\theta)$. Here, the posterior $p(\theta / D)$ is a probability density function (PDF) of model parameters θ given the data, the likelihood $p(D / \theta)$ is the PDF of the data given the model parameters θ , and the prior $p(\theta)$ is a PDF reflecting a priori beliefs about values model parameters can assume. Note that the posterior is proportional to the product of two PDFs of two distinct random variables, D and θ . By contrast, the posterior means and variances specified in Eq. 1-3 result from a mere multiplication of two PDFs of a single random variable. Without further explanation, this appears like a misapplication of Bayes rule. Greater mathematical rigor is needed to clarify how Bayes’ rule was applied, including explicit definitions of model parameters (θ), data (D), and the likelihood model. Given these conceptual and technical shortcomings - and considering the analysis is not central to the paper’s primary conclusions - I suggest removing this section.

We thank the reviewer for this important observation and agree that our initial phrasing may have led to confusion. In our formulation, we did not attempt to derive a full posterior over model parameters but rather implemented an analytical solution to the posterior over the sensory estimate (pain intensity) using a conjugate prior formulation under the assumption of normally distributed priors and likelihoods. The equations presented in Eq. 1–3 correspond to the analytical form of the posterior mean and variance resulting from Bayesian integration over stimulus intensity, not over model parameters. Accordingly, the model parameters (e.g., pshift, pprecision, poffset) were estimated using variational Bayesian inference over trials, not derived via Bayes’ theorem in Eq. 1–3. We have now clarified this distinction and updated the text to reflect that Eq.

1–3 describe the application of normal-normal conjugate inference on a cognitive model of perceived intensity, rather than an inferential Bayesian treatment of model parameter distributions.

In our framework, $\mu_{\text{posterior}}$ and $\sigma^2_{\text{posterior}}$ do not reflect a posterior over model parameters θ , but the mean and variance of the *subjective percept*, computed through Bayesian integration of a prior belief about pain intensity (based on the cue) and the sensory evidence (likelihood). Accordingly, the variables in our equations correspond to hypothetical internal distributions over pain intensity, not statistical distributions over data or parameters. We regret any confusion and have clarified this distinction in the revised manuscript at several points.

Although modeling is not central to the primary findings, it provides a principled framework to formally contrast alternative mechanistic hypotheses (shift vs. precision modulation), and to explore whether temporal aspects of expectation affect pain perception through distinct computational routes (i.e. via changes in intensity or precision). While Weber’s law originally refers to perception of time intervals, we here adopt its broader implication: that internal uncertainty increases with longer intervals. Applied to expectation, we hypothesize that longer cue-stimulus delays reduce the influence of the prior due to increased uncertainty in applying temporally distant cues, i.e. they would be “less reliable” – based on the larger uncertainty induced by Weber’s law. We believe this contributes to the broader theoretical context and therefore retained the section in revised form. (See lines 234ff., 241, 249ff., 272ff., 288ff., 296ff., 304ff., 332ff., 580ff., 832ff., 857, 865ff., 889ff., 928ff.)

- The discussion claims that “previous EEG pain studies have consistently shown that cue-induced expectation effects do not modulate EEG signals during pain stimulation ...”. This is not true. For instance, Bott et al. (2023, Science Advances) demonstrated that cue-induced expectations modulate interregional connectivity during pain stimulation. The study’s exclusive focus on signal power should be noted as a limitation.

We thank the reviewer for highlighting recent EEG work. We now explicitly acknowledge that while our EEG analysis focused on signal power, prior studies have shown expectation effects in connectivity and phase-based metrics during pain stimulation. We also added this point in the limitations. (See lines 600ff., 636ff.)

- The discussion states that the paradigm “eliminates the confound of expected waiting times and actual waiting times”, but this requires clarification. As I understand it, by presenting cues about the waiting time, expectations about waiting times and thus their confound with actual waiting times is introduced in the first place. However, it is also true that the paradigm allows for the disentanglement of the two factors.

We agree and expanded the discussion to clarify that while our design introduces expectations via cues, it also allows for their dissociation from actual delays, which is the key advantage for studying predictive timing with the present design. (See lines 549ff.)

- The description of the repeated measure ANOVA in the results and methods sections is unclear (“One-way interaction” is not standard terminology. What are the ANOVA factors? How was the prediction error effect tested specifically? Why is the data summarized in a 2 x 35 x 9 “matrix”? While

expected and actual durations are categorical variables with three levels, the phrase “measured response [is modeled] with a simple linear function of stimulus latency” implies these variables are treated as continuous). Providing the code for the analyses would greatly facilitate their understanding (see comment below).

We thank the reviewer for requesting clarification. We have now restructured the Results and Methods sections to clearly list all ANOVA factors: stimulus modality (heat/cold), expected delay (cue-based), actual delay (stimulus latency), and prediction error (absolute difference). We no longer use “one-way interaction” terminology and provide the MATLAB code. (See lines 157f., 807ff.)

- To promote transparency and reproducibility, the analysis code and data should be made publicly available and easily accessible.

We agree and had already made data available at OSF (as mentioned in the original manuscript). We have now added the updated scripts. (See lines 969ff.)

- A priori sample size calculations should be provided. Note that lower power entails a lower likelihood of positive effects to reflect true effects.

We based our sample size on previously observed behavioral effect sizes from our related study (Strube et al., 2023) and added a corresponding paragraph in the Methods section. (See lines 655ff.)

Minor concerns

- Are predictive coding principles truly fundamental to all brain processes, or primarily relevant to perception-related processes? (Lines 43-44)

This is a heavily debated question, and our study was not designed to substantially add to this discussion. We therefore have rephrased this section accordingly. We now discuss Predictive Coding as a possibility, for perception-(and-pain-) related processes without claiming universality across all brain functions. (See lines 39ff.)

- Timing may be a critical factor in clinical pain contexts, but it is unclear how this specifically motivates the present study (Lines 61-68).

We agree and have removed the clinical framing from the introduction.

- Why were non-painful cold stimuli used instead of non-painful warm stimuli?

We have added an explanation to the Methods section stating that cold stimuli were chosen over warm stimuli due to pilot results showing stronger perceptual clarity and lower habituation effects. Warm stimuli near the neutral threshold (32–34 °C) were perceived inconsistently (i.e. some trials were missed), whereas cold stimuli (20.5 °C) showed clearer responses across trials. (672ff.)

- The coloring of the Table in Figure 1a (color intensity ~ actual latency) is inconsistent with that in Figure 1b (color intensity ~ expected latency).

Thank you! We have corrected the color mapping in Figure 1a to align with Figure 1b

- Line 194: The phrase "...modulate sensory perception in a way that it is experienced as more intense" is awkward and suggests unintended links to consciousness research ("experience of perception").

We agree and have revised the sentence for clarity. (203f.)

- The bar graphs in Figures 3a and 3b reveal little about the data, and the lower cut-offs seem arbitrary. Consider using box plots or raincloud plots instead.

We appreciate the suggestion and have replaced the original bar plots with raincloud plots in Figures 3a and 3b to better visualize the data distribution.

- Line 293: Replace "time points" with "time intervals"?

We have replaced "time points" with "time intervals". (See line 341)

- For the assessment of stimulation effects on EEG, why were only correctly cued trials used?

We used only correctly cued trials in the EEG analysis to avoid confounding effects of prediction errors on neural responses. (See lines 362ff.)

Reviewer #3 (Remarks to the Author):

Key Results

This study operationalises a predictive timing paradigm to examine how actual and expected pain stimulus latency influence subjective pain intensity ratings across hot and 20.5-degree (cold) stimulus modalities. By implementing computational Bayesian modelling, the authors investigate the role of prior expectations and precision in shaping pain perception within a predictive coding framework, particularly in relation to the dread effect - the idea that prolonged anticipation amplifies pain perception. EEG data were used to explore the neural signatures of probabilistic inference processes. The results indicate that actual stimulus latency does not significantly affect ratings, but temporal expectations do, such that delayed stimulation was rated as more painful than instant stimulation. Bayesian model selection supported a prior expectation intensity shift model over a prior expectation precision modulation model, suggesting that the content of top-down predictions exerts a stronger influence on ratings than their certainty. The same pattern was observed for the 20.5 degree ("cold") stimulation suggesting that the effect of temporal expectations is not pain-specific.

The key EEG data showed that temporal expectations modulated alpha-to-beta frequencies at cue presentation, but not at stimulus delivery, supporting the argument that they bias prior expectations rather than the processing of ascending nociceptive signal. Temporal prediction errors were expressed in the beta-to-gamma frequency range.

The key EEG data showed that temporal expectations modulated alpha-to-beta frequencies at cue presentation, with a crossed interaction where activity increased for heat and decreased for cold. No significant modulation of alpha-to-beta power was observed at stimulus delivery, supporting the argument that they bias prior expectations rather than the processing of ascending nociceptive signal. Temporal prediction errors were expressed in the beta-to-gamma frequency range,

particularly in the gamma range, during the second half of stimulus presentation.

Validity

The experiment offers a valid test of the hypotheses. The rationale is very clear and followed up well by comprehensive analyses, which closely follows the stated rationale. The behavioural modelling is particularly impressive, and all results are examined thoroughly, and interpreted fairly.

A few minor corrections can help increase the validity:

- The 'cold' stimulus is referred to as 'aversive' and 'unpleasant'. Since 20 degrees is not a low temperature this designation should be removed unless it can be supported by evidence.

We agree and removed the labeling of 20.5°C cold stimuli as “aversive” or “unpleasant.” We now simply describe them as “non-painful cold.”

- The main effect of stimulus intensity on the signal at stimulus delivery (line 368 details the statistical model) is not reported – was there any difference between 'hot' and 'cold'?

We now report the results of the main effect of modality (cold vs heat) in the Supplementary Materials (Supplementary Figure 2). (See lines 361ff.)

- The maximum cue-stimulus interval of 4 seconds in this study may not engage the prolonged affective anticipation necessary to elicit dread-related modulation of pain perception. The null effect of actual stimulus latency may therefore reflect an insufficient experimental latency window, a possibility which should be acknowledged as a caveat.

In our limitations sections, we now discuss that our 4-second maximum delay may not reflect the temporal range of “true” dread-related affective buildup. (See lines 633ff.)

- The modelling assumes that the mean and precision of prior expectations are static across trials of equal expectancy conditions. A more biologically plausible implementation would involve a hierarchical precision estimation process, where precision is continuously updated in response to ongoing sensory evidence and past prediction error rather than being pre-determined by task conditions. The authors may wish to consider the assumption as a potential limitation.

We agree that our model assumes fixed priors within conditions and now explicitly state this as a limitation. We mention that a Kalman-filter or hierarchical Bayesian implementation could model trial-by-trial precision updates and may be a future direction. (See lines 640ff.)

Significance

How the effect of dread can be operationalised within predictive processing mechanisms, and how it is implemented computationally and neutrally, is a fascinating topic and one that the authors have done an excellent job addressing. The study contributes to our understanding of predictive coding in pain perception by addressing the topic of temporal predictions and prediction errors which has been relatively neglected.

Data and Methodology

The method and analysis are professionally executed.

Analytical Approach

The analysis was executed professionally. Given the core emphasis on Bayesian modelling and the theoretical significance of some null effects we were curious whether the authors might consider employing Bayesian rather than frequentist statistics.

Suggested Improvements

- The authors should report how many trials were included in each condition/participant

We now provide the number of trials per condition and participant in the Supplement (Supplemental Table 1 & 2). (See lines 752f.)

- The behavioural analysis reports interactions with stimulus intensity (hot/cold); it should also unpack/explain this interaction.

We thank the reviewer for pointing this out. Upon revisiting our analysis, we realized that a mistake had occurred due to incorrect centering of the factors. In particular, the modality factor (hot/cold) was not properly centered [1 1 1 1 1 1 1 1 1 0 0 0 0 0 0 0] instead of [1 1 1 1 1 1 1 1 1 1 -1 -1 -1 -1 -1 -1]. After correcting this issue and rerunning the repeated-measures ANOVA, we found that the interaction effect is no longer statistically significant. As already outlined in the previous version of the manuscript, this interaction was not central to our results. We have now updated the manuscript and OSF files accordingly and now report the corrected version of the analysis without a significant interaction effect. Nevertheless, we still report the results of the post-hoc tests for completeness and transparency. We apologize for the mistake and thank the reviewer for bringing this to our attention. (See lines 157ff.)

- Comparing hot/cold separately against catch trials is helpful, but titling Figure 5 'main effect' could be misleading, because this term is typically reserved to the results of a factorial analysis. That figure could improve by including the main effect from the F test in line 368.

We have renamed the figures and adjusted captions to avoid confusion. (See lines 355ff.)

Clarity and context

The manuscript is well-structured and provides a clear account of the experimental approach and findings. However, the categorical treatment of thermos-sensation introduces potential conceptual and statistical ambiguities that are not explicitly acknowledged.

We now explicitly acknowledge this limitation in the revised manuscript, noting that future work could explore continuous modeling of thermal intensity across a broader stimulus range. (See lines 629ff.)

References

The manuscript cites relevant literature on predictive coding, Bayesian inference, and pain perception,

but the discussion on precision weighting could be better contextualised within the broader hierarchical inference literature. Specifically, research on error-driven precision updates and adaptive inference mechanisms should be referenced to clarify how the study's approach differs from models that incorporate trial-by-trial learning of precision estimates.

Thank you for this helpful comment. We now clarify in the Limitations that our approach assumes stable prior expectations within blocks, due to the fully balanced block design. We now mention that models with trial-by-trial learning would be a potential future direction and use appropriate references. (See lines 638ff.)

Reviewer #4 (Remarks to the Author):

We thank the co-reviewer for their contribution.

REVIEWER COMMENTS

Reviewer #1 (Remarks to the Author):

I am satisfied with the authors' revisions and have no additional comments.

We thank the reviewer for their positive evaluation and are pleased that they are satisfied with our revisions.

Reviewer #2 (Remarks to the Author):

With their revisions, the authors have substantially improved the manuscript. Below are my remaining comments.

- The revised framing and reduced emphasis on Predictive Coding are now more appropriate.

We are glad that the revised framing and the reduced emphasis on Predictive Coding are now more appropriate.

- The explanation of the Bayesian methodology is clearer, and the provided references are helpful. However, the manuscript would benefit from a more rigorous presentation of the approach, e.g., in the supplementary materials. Specifically, explicitly stating the prior, likelihood, and posterior densities (along the lines of Körding and Wolpert, 2004) together with their respective parameters (means and standard deviations) would aid understanding.

We thank the reviewer for this comment. In the revised Supplementary Materials, we now provide a more rigorous presentation of our Bayesian modeling approach. To illustrate interindividual variability and robustness, we include summary figures showing distributions of parameter estimates, as well as their correlations with the behavioral expectation effect. Together, these additions address the reviewer's request for a more transparent and precise description of the Bayesian framework. We think that this significantly improved our manuscript. (See Supplementary Information; see Supplementary Figure 1)

Moreover, in my opinion, the following statement is problematic: “, where priors and likelihoods represent internal beliefs, not random variables in a statistical inference framework”. Priors and likelihoods, are, by definition, associated with random variables. It is these random variables which are used in conjunction with Bayes theorem to model internal beliefs and the subjective percept.

We thank the reviewer for this important clarification. We agree that our original formulation was confusing and misleading. In the revised manuscript, we have removed this statement. (See line 903)

Lastly, in line 924, the procedure for estimating the parameters should be described more precisely. E.g., explicitly list the parameters which are estimated and state the optimization criterion used.

We thank the reviewer for this helpful suggestion. We now explicitly state that parameter estimation is achieved by a variational approach to approximate Bayesian inference and based on the minimization of variational free energy as implemented in variational Bayes and the VBA Toolbox (<https://mbb-team.github.io/VBA-toolbox/>). We have also listed the estimated parameters (p_{shift} , $p_{\text{precision}}$ and p_{offset}) in the revised manuscript. (See lines 987ff.)

- Regarding the potential confound of expected and actual waiting times, the authors have added an explanatory paragraph, which makes their point clearer (lines 553 - 559). However, in its current form, the added text appears somewhat unconnected with the rest of the explanation. I suggest explicitly stating that the absence of expectation cues does not imply the absence of expectations, if this is what the authors mean.

*We thank the reviewer for the comment and apologize that we were not clearer in our first revision.*
*In the revised manuscript, we have clarified this point and rephrased the paragraph accordingly.*
*We now explicitly state that the absence of expectation cues does not imply the absence of*
*expectations. We explain that expectations without cues are gradually shaped by elapsed time,*
*whereas in our paradigm, expectations are formed immediately based on the cue and therefore*
*disentangled from actual waiting times. (See lines 577ff.)*

- The description of the statistical model has improved. However, should it not be “ratings were
averaged within each combination of” rather than “across all combinations of” (line 807)?

*We thank the reviewer for this careful observation. In the revised manuscript, we have replaced the*
*repeated-measures ANOVA with a single-trial Linear Mixed-Effects (LME) approach. This model*
*does not rely on averaging across or within combinations of conditions, but instead considers all*
*predictors simultaneously at the trial level, with subject-specific random intercepts and slopes. The*
*full model specification and output are now provided in the Supplementary Information for*
*transparency. (See Supplementary Information; see lines 153ff., 205ff., 872ff.)*

- There is no code available under the provided link. Providing the study's code and would enhance its
transparency and reproducibility.

*The OSF link provided in the manuscript is active, and all analysis code is available there. The*
*scripts are located in the “Files” section of the repository, organized into several subfolders labeled*
*“Analysis Code.” We apologize if accessing the code may have been unintuitive; we have tested the*
*link and confirmed that all files can be retrieved. (See lines 1071f.)*

Reviewer #3 (Remarks to the Author):

The available evidence from prior research suggests that participants tend to anticipate greater pain
when it's expected to occur after a delay—possibly accounting for the so-called “dread effect,” in
which people would rather experience pain sooner than later. It's intriguing to consider whether this
very expectation might amplify the subjective intensity of pain, and it's worth exploring how
expectations operate—whether by shifting the mean or altering the variance. Since the behavioral
finding runs counter to intuition, the entire manuscript deserves a close, critical read. After reviewing
the revised version, several significant concerns emerged.

Major Comments: Interpretation and Modeling

The mean shift model strikes me as a relatively straightforward application of predictive coding
principles to these findings. Given that participants report more intense pain when they anticipate a
longer delay—and considering that a prior expectation with a higher mean would naturally lead to a
higher posterior—the idea of modeling this as a shift in the mean seems both logical and grounded. It
offers a solid theoretical framework, especially valuable if the goal is to test alternative explanations of
the data.

*We thank the reviewer for this positive evaluation of our work.*

That said, I'm increasingly unconvinced that the precision-shift model offers a meaningful
comparison. While I understand the premise that priors might become less precise with longer
expected delays, it's not clear how this imprecision could lead to a mean-shifted posterior. The authors
could strengthen their case by more convincingly explaining why this model represents a plausible
mechanism, rather than simply re-describing the ANOVA results through a more complex lens.

*We thank the reviewer for this important comment. We agree that our initial description of the*
*precision-shift model was too brief and may not have sufficiently conveyed its theoretical*
*motivation. In the revised manuscript, we now explain in more detail why the modulation of prior*
*precision represents a plausible mechanism. Specifically, when temporal expectations become less*
*precise, the sensory input receives relatively greater weight in the Bayesian integration process. If*

*the actual sensory input is more painful than the expected value, this reweighting results in a*
*posterior shifted toward higher pain intensity. Crucially, this account is not redundant with the*
*mean-shift model: while both mechanisms predict a higher posterior mean, the precision-shift*
*model additionally predicts a decrease in posterior precision (i.e., greater variability in perception).*

*In this illustrative figure (reproduced from Strube, 2022), two scenarios are shown: a prior shift*
*(purple, top) and a likelihood precision modulation (green, bottom). In both cases, the posterior*
*distribution (blue) is shifted toward lower VAS values, but the underlying mechanisms differ.*

- *With a prior shift, the posterior mean decreases because the prior itself is shifted.*
 - *With a likelihood precision modulation, the likelihood becomes less precise, reducing its weight in the Bayesian integration. As a result, the posterior is drawn more strongly toward the relatively more precise prior.*

*Importantly, the posterior precision depends on the precision of both prior and likelihood. A prior*
*shift only affects the posterior mean, whereas a likelihood precision modulation affects both the*
*posterior mean and posterior precision, leading to increased variability in the perceptual estimate.*

*We have now clarified this distinction. In the Supplementary Information, we additionally provide*
*an illustrative example of a scenario in which the precision-modulation model outperforms*
*alternative accounts, yielding not only a higher posterior mean but also a lower posterior precision*
*by this mechanism. (See Supplementary Information; see lines 253ff., 322ff.)*

The discussion of temporal uncertainty should also be contextualized with prior work, particularly
Clark et al. (2008), who demonstrated that elevated pain ratings weren't attributable to temporal
uncertainty alone.

*We thank the reviewer for pointing this out. Please refer to our response below, where we address*
*the suggestions about the contextualization of our results within the literature which examined the*
*effects of delay on pain.*

Additionally, the authors suggest their model supports the idea that “cognitive/affective processes alter
the valence of late-expected stimuli,” but given the lack of an interaction between expectation and
modality (line 173), I’m unsure why a valence-based explanation is justified.

*We agree that our earlier interpretation was too speculative and we have now removed this passage.*

The EEG findings are informative in showing that expectations influence brain activity at the moment
the cue is processed but not when the stimulation is processed. This interesting difference between the
effect of expectations in two time points should be backed up by including cue and stimulus in the
same analysis. However, the broader mechanism underlying the key behavioral effect remains unclear.

***We thank the reviewer for this suggestion. We agree that it is important to compare cue-related and***
***stimulus-related effects; however, within the framework of cluster-based permutation testing,***
***concatenating cue and stimulus periods into a single analysis is not feasible. This approach***
***requires temporal alignment across conditions, which is not possible here because the stimulus***
***phase follows different waiting times depending on the cue condition. Concatenating the two phases***
***would therefore violate the assumptions of the test and could yield uninterpretable results. Instead,***
***we analyzed cue-locked and stimulus-locked data separately, which allows us to capture the***
***temporally specific effects of expectations at the moment the predictive cue is processed and at the***
***time of stimulation.***

Major Comment: Manipulation Check

The experiment uses a factorial design, with modality as a critical variable. This factor is included in
the behavioral analysis and again in analyzing temporal delay (line 412). But oddly, it's missing as an
independent variable in the immediately preceding EEG analysis. Given the study design, the
appropriate test for whether EEG responses differ by modality would be a direct comparison of heat
vs. non-painful stimulation, with that contrast featured prominently in the main text (rather than being
relegated to the supplementary material).

More troubling, the manipulation check fails to replicate the findings the authors committed to
replicating—namely, the pattern identified in Ploner et al.'s (2017) review: increased theta and
decreased alpha-beta. While this pattern does emerge in the heat vs. catch trial comparison, that
analysis is confounded by the mere presence versus absence of stimulation. The cleaner test—heat vs.
control stimulation—does not replicate this expected result, even though it's the more appropriate
comparison for a factorial design involving modality.

***We thank the reviewer for this important point. In the revised manuscript, we now include the heat***
***vs. cold control contrast in the main Results section (rather than only in the Supplement). We also***
***updated the interpretation of our findings accordingly: while the heat vs. catch comparison***
***reproduces canonical EEG patterns of pain (theta increase, alpha–beta decrease), the heat vs. cold***
***contrast shows a different profile, with increased theta–alpha and beta–gamma activity for pain.***
***However, this is not at odds with previous findings (e.g. Ploner 2017) as we here compare heat to***
***cold stimuli. We have clarified this distinction in the Results and now also explicitly discuss it in the***
***Discussion to highlight both the convergence with and the divergence from canonical pain-related***
***oscillatory responses. (See lines 356ff., 399ff., 411ff., 644ff.)***

Major Comment: Reporting Statistics

• Lines 157, 167, and 814 suggest that just one ANOVA was conducted, but from lines 805–821, it's
still unclear exactly which statistical tests were run. Were all main effects entered simultaneously? If
so, the design would likely be unbalanced (i.e., unequal numbers of trials across some of the 9
conditions). How was this addressed?

• Including both prediction error and latency in the same analysis seems problematic, since the PE
calculation combines different conditions (e.g., high PE for late onset/0s cue and early onset/4s cue).
Did the authors run multiple two-factor models?

• Lines 178 and 183 suggest that real and expected delay were entered together, yet the degrees of
freedom reported for their interaction with modality are (1,34), not (2,34) as would be expected with 2
modalities and 3 levels of latency. This discrepancy needs clarification.

• Results from normality tests and information on how violations of sphericity were handled should be
reported.

*We thank the reviewer for these comments. To address the concerns, we re-analysed the data using*
*a more comprehensive single-trial Linear Mixed-Effects (LME) models (see Methods &*
*Supplement). These models include all predictors simultaneously at the trial level and explicitly*
*model subject-level random intercepts and slopes. This approach: (1) accommodates the*
*unbalanced trial counts without excluding any subjects, (2) allows Expectation and Prediction*
*Error (PE) to be included as continuous predictors so that their unique contributions are estimated,*
*(3) does not rely on the sphericity assumption since the within-subject covariance is explicitly*
*modelled via random effects, and (4) provides appropriate degrees of freedom and p-values via*
*mixed-model approximations.*

*We ran two LME variants (PE and PE_Stim), and in both cases the pattern of results was*
*unchanged relative to the repeated-measures ANOVA: there was a robust main effect of Modality*
*and a significant effect of Temporal Expectation, whereas Latency and PE did not significantly*
*modulate ratings, and no interactions with Modality reached significance. (See lines 153ff., 872ff.)*

Minor Comments

• Line 210 contains a typo: “This is at odds with interpretations explaining increased pain by higher,
instead,” – sentence is incomplete or awkwardly phrased.

*We thank the reviewer for pointing out this typo. We have revised the sentence. It now reads: “This*
*is at odds with interpretations explaining increased pain by higher arousal; instead, this suggests*
*that the somatosensory expectation itself can drive this effect in both painful heat and non-painful*
*cold conditions.” (See lines 200ff.)*

• Figure 1: For “probability at stimulation,” the catch trials should list “N/A” instead of “100%,” since
no stimulation occurs.

*We thank the reviewer for pointing this out. We have corrected Figure 1 accordingly: for catch*
*trials, “probability at stimulation” is now indicated as “N/A” instead of “100%.” (See Figure 1)*

• EEG trial counts: Some condition counts are reported as fewer than 10 trials. The authors should
reassure readers that each condition included in Figure 8 had a sufficient number of trials for robust
analysis—typically more than 10, and ideally over 20, as per common practice (see Supplementary
Table 1).

*We thank the reviewer for highlighting this important point. We have now included detailed trial*
*counts for all conditions shown in Figure 8 in the Supplementary Information (Supplementary*
*Table 3 & 4), to reassure readers that the analyses are based on sufficient data - typically more than*
*10 trials per condition, and often substantially more.*

*For the Prediction Error (PE) analysis, the minimum number of trials in the high-PE condition was*
*8 for a few participants. While this number is somewhat lower than the commonly cited threshold of*
*10 trials, we note that it is still within the range considered acceptable for EEG analyses (see e.g.,*
*Olvet & Hajcak, 2009, for comparable trial counts in ERP studies). Based on this reasoning, and*
*given the overall consistency across participants, we decided to retain the full dataset for the main*
*PE analysis reported in the manuscript.*

*At the same time, to address the reviewer’s concern and to demonstrate robustness, we performed*
*an additional analysis of the PE time-frequency effect including only participants with ≥ 10 trials in*
*the high-PE condition. The results of this restricted analysis were fully consistent with the main*
*analysis and are now reported in the Supplementary Information. This convergence confirms that*
*our conclusions are not driven by a small subset of participants with lower trial counts. (See*
*Supplementary Figure 3; see Supplementary Table 3,4)*

• While the paper rightly emphasizes the novel design element—separating expectation from actual
delay via the cue—it would be valuable to relate the EEG results back to prior M/EEG studies that

examined actual delay. That comparison would help clarify whether earlier findings were truly driven
by expectations or were confounded by the delay itself.

*We thank the reviewer for this helpful suggestion. We now explicitly discuss Clark et al. (2008) and*
*Hauck et al. (2007) in the revised Discussion. The null effect in Clark et al. for short delays (3 vs. 6*
*s) aligns with our finding of no effect for actual delays (0–4 s), while their P300 modulations by*
*unpredictability parallel our EEG prediction-error effects without behavioral changes. Findings of*
*Hauck et al. (2008) of higher pain with longer delays are reinterpreted as confounded by higher*
*pain probabilities, whereas our paradigm separates expected from actual delay and controls for*
*probability. (See lines 588ff.)*

*We believe that the current version of the manuscript fully addresses all major and minor comments*
*raised by the reviewers. Importantly, the additional analyses (e.g., LME, Bayesian modeling*
*clarifications, EEG contrasts) converge with our original findings and strengthen the robustness*
*and transparency of the study. We therefore hope the manuscript is now suitable for acceptance in*
*Nature Communications.*

REVIEWER COMMENTS

Reviewer #3 (Remarks to the Author):

The revised manuscript represents a clear improvement aligned with reviewer feedback, but this
increased clarity raises important concerns about the modelling.

On temporal uncertainty, they now reference Clark et al. (2008) and position their results in relation to
that literature. They also further contextualise their EEG findings relative to Ploner et al. (2017),
clarifying the basis for comparison in that review and discussing why a heat/cold contrast can yield a
different profile without contradiction.

The revisions provide clearer language and interpretations of results and a cleaner distinction between
actual and expected latency, making the central arguments easier to follow. The revision explains how
separating expected from actual delay addresses prior designs that bundled these factors and clarifies
why short-delay nulls can arise under such confounding.

The authors have added clarifications to how their precision-modulation account yields a mean shift in
the posterior conditional on a prior $<$ likelihood asymmetry, and helpfully further specified the
assumptions accountable for its behaviour, making explicit the directional assumptions used in their
illustrative figures.

However, the implementation of the “expectation precision modulation model” is still problematic.
The model makes a critical assumption that the expected mean pain (the prior mean) is lower than the
sensory likelihood (via p_{offset}). The authors do not discuss this in any length, but this assumption is
necessary to enable reduced prior precision to shift the posterior upward for longer expected delays.
While it is entirely possible that sometimes expected pain would be lower, in other times it may be
higher, exceeding input across trials. It is not obvious why Bayesian updating would not centre
expectations exactly on the intensity of delivered pain, especially considering that uniform stimulation
intensities are used (heat/cold).

The “expectation intensity shift model” directly shifts the prior mean upward with delay (via p_{shift})
to fit observed higher ratings, but provides no cognitive or theoretical basis for why expected latency
would systematically inflate intensity expectations. Precision modulation has firm support in Bayesian
perception accounts, where longer expected delays diminish prior precision under temporal
uncertainty. The mean-shift mechanism lacks similar grounding, which makes it read as an add-on that
enables a dread-like effect, but not derived from a specified generative link between delay cues and
intensity expectations. Without an explicit rationale that elevates it to a general structural (dread-like)
feature in line with the principal computational frameworks, its mechanistic and epistemic basis
remains uncertain, and is further challenged by the same monotonic effect in the non-painful cold
condition.

These issues seem to stem from an underlying epistemic oversight. Both models incorporate
assumptions that presuppose that the dread effect exists in order to illustrate it in the data, rather than
deriving a model structure from independent Bayesian principles that naturally reproduces it in line
with predictive and Bayesian perception accounts. This arguably makes the modelling tautological, in
that it is designed to fit behaviours while assuming the effect, rather than to fit the effect assuming a
theory of cognition.

As a result, the modelling appears more like parameterisations describing data structure, rather than
explaining why that structure occurs. It is recommended that the authors either 1) reframe the focus of
the modelling from explanatory to descriptive, or 2) develop a strong rationale that can justify the
assumption of a general prior-mean-shift mechanism in the context of Bayesian perception, and/or the
assumption that a general mechanism may deflate pain expectations, moving the prior mean below the
likelihood mean, so that the latency-dependent precision reduction may push posteriors upwards.

These constitute plausible challenges to the authors’ interpretation with respect to epistemic
parsimony, with implications for hypothesis testing. An acknowledgement of this theoretical difficulty
in the main text would hedge against strong critiques of the Bayesian model selection results on
theoretical grounds.

*We thank the reviewer for the thoughtful assessment of our modelling approach. We agree that our*
*previous version unintentionally risked framing the modelling as mechanistic, even though the*
*underlying assumptions necessarily make it illustrative rather than explanatory. In the revised*
*manuscript, we therefore position the modelling explicitly as exploratory and descriptive, retaining*
*only a short summary in the main text. All model specifications, assumptions, parameters,*
*simulations, and Bayesian model comparisons have been moved to the Supplementary Information,*
*where they can be consulted by readers interested in computational perspectives on pain. We believe*
*this restructuring prevents any over-interpretation while preserving transparency and the potential*
*value of the modelling as an illustration of a possible mechanism. In particular, this could stimulate*
*future research that directly addresses the assumptions we made in the context of our model.*

*We are genuinely grateful for this comment. It led to a clearer presentation of the modelling*
*approach, and thus improved the conceptual precision of the manuscript. (See Supplementary*
*Information; see line 22; see lines 202ff.; see line 516f.)*
